# TimeBase: The Power of Minimalism in Efficient Long-term Time Series Forecasting

Qihe Huang [1] Zhengyang Zhou [1,2,3] Kuo Yang [1] Zhongchao Yi [1] Xu Wang [1,2] Yang Wang [1,2]

## Abstract

Long-term time series forecasting (LTSF) has traditionally relied on large parameters to capture extended temporal dependencies, resulting in substantial computational costs and inefficiencies in both memory usage and processing time. However, time series data, unlike high-dimensional images or text, often exhibit temporal pattern similarity and low-rank structures, especially in long-term horizons. By leveraging this structure, models can be guided to focus on more essential, concise temporal data, improving both accuracy and computational efficiency. In this paper, we introduce TimeBase, an ultra-lightweight network to harness the power of minimalism in LTSF. TimeBase 1) extracts core basis temporal components and 2) transforms traditional point-level forecasting into efficient segment-level forecasting, achieving optimal utilization of both data and parameters. Extensive experiments on diverse real-world datasets show that TimeBase achieves remarkable efficiency and secures competitive forecasting performance. Additionally, TimeBase can also serve as a very effective plug-and-play complexity reducer for any patch-based forecasting models. Code is available at https://github.com/hqh0728/TimeBase.

## 1. Introduction

Long-term time series forecasting (LTSF) has been studied with significant interest in various domains, ranging from energy management, traffic accident preservation, and extreme disaster warning. With the rapid advancement of deep learning, an increasing number of models have been proposed (Qiu et al., 2024; Zheng et al., 2023; Ang et al., 2023; Chen et al., 2023; Wu et al., 2021; Zhou et al., 2023a; Wang et al., 2023b; Lin et al., 2023a; Ding et al., 2018; Chen et al., 2018; Wang et al., 2024c; Chen et al., 2019; Wang et al., 2023a; Huang et al., 2024c; Zhou et al., 2023b; Wang et al., 2024b; Lin et al., 2024a; Miao et al., 2025; Yi et al., 2024), including MLP-based (Liu et al., 2022; Huang et al., 2024a; Miao et al., 2024), RNN-based (Lin et al., 2023b), and Transformer-based (Liu et al., 2021b; Zhang and Yan, 2023), approaches, all of which employ thousands to millions of parameters to capture long-range dependencies and forecast future outcomes.

Generally, a higher number of parameters increases the model capacity, which can lead to better predictive performance (Zhou et al., 2023c; Zhao et al., 2024). In the fields of computer vision (CV) and natural language processing (NLP), large models have achieved significant success (He et al., 2016; Liu et al., 2023). For instance, Vision Transformers (ViT) (Dosovitskiy, 2020) have demonstrated outstanding capabilities in image recognition, while large language models (LLM) have made breakthrough advances across various language tasks (Devlin, 2018; Radford et al., 2019). Recently, large models are being explored for LTSF to capture complex temporal patterns and long-range dependencies (Cheng et al., 2024; Liu et al., 2025a; 2024d; 2025b). For example, some LLM-based forecasting methods are proposed with tens of billions of parameters (Jin et al., 2023). However, despite their impressive performance on specific forecasting tasks, these models suffer from high computational costs and resource-intensive requirements.

In fact, images and text, as high-dimensional data, contain multiple dependencies and complex underlying physical rules (Liu et al., 2021a), which necessitate the use of more parameters to model their rich semantic structures. However, as shown in Figure 1(a), one-dimensional time series data is typically much more regular, exhibiting obvious tem-

[1]University of Science and Technology of China (USTC), Hefei, China [2]Suzhou Institute for Advanced Research, USTC, Suzhou, China [3]State Key Laboratory of Resources and Environmental Information System, Beijing, China. Qihe Huang <hqh@mail.ustc.edu.cn>. Kuo Yang <yangkuo@mail.ustc.edu.cn>. Zhongchao Yi <zhongchaoyi@mail.ustc.edu.cn>. Xu Wang <wx309@ustc.edu.cn>. Correspondence to: Zhengyang Zhou <zzy0929@ustc.edu.cn>, Yang Wang <angyan@ustc.edu.cn>.

*Proceedings of the 42$^{nd}$ International Conference on Machine Learning*, Vancouver, Canada. PMLR 267, 2025. Copyright 2025 by the author(s).

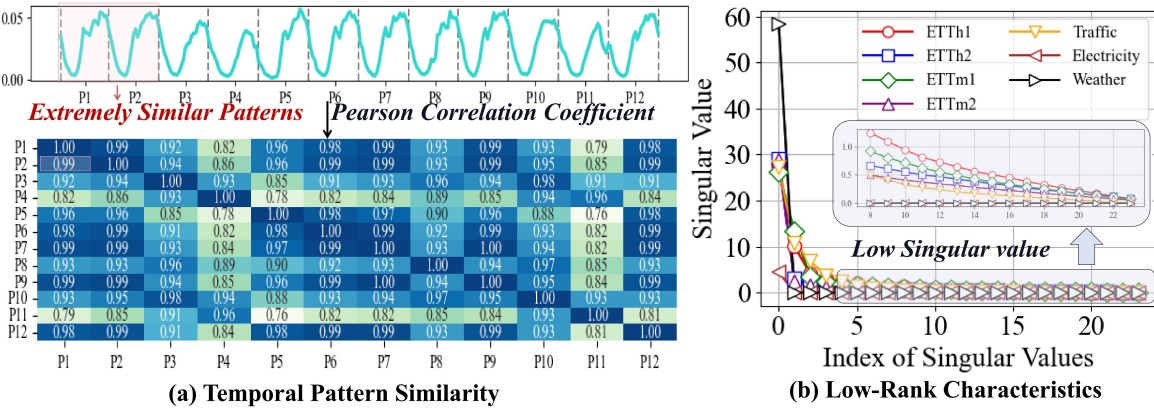

*Figure 1.* Illustration of temporal pattern similarity and approximate low-rank characteristics in long-term time series.

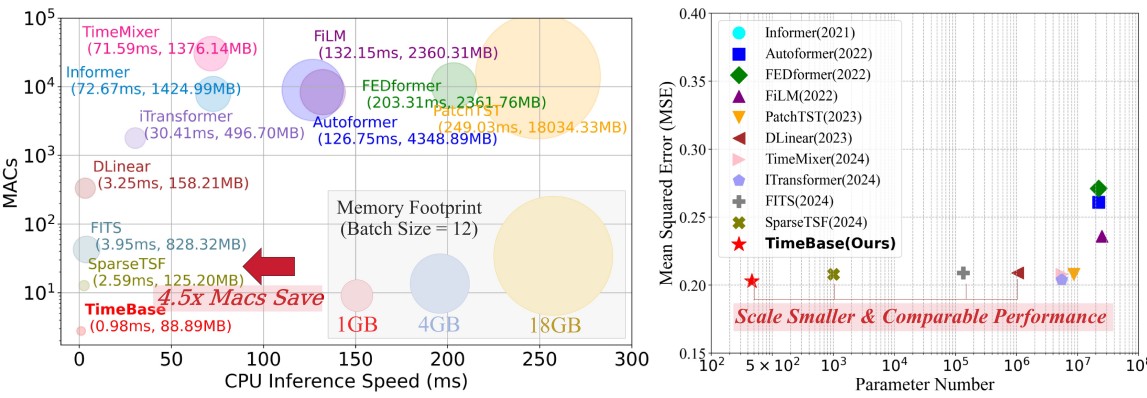

*Figure 2.* Comparison of forecasting performance and model efficiency in terms of MSE, MACs, parameters, memory usage, and CPU inference speed for 720-horizon forecasting in Electricity.

poral patterns. Moreover, in long-term time series data, this regularity can even manifest as low-rank characteristics (Liu et al., 2012), which can be reflected through singular value decomposition as shown in Figure 1(b), indicating that there is a considerable amount of redundant information (Jones and Brelsford, 1967; Hochreiter and Schmidhuber, 1997). This raises an important question: Is it truly necessary to employ such a large number of parameters to learn these regular time series patterns (Tan et al., 2024; Zuo et al., 2024; Khayati et al., 2024)?

In this study, we design an extremely lightweight time series forecasting network, TimeBase, which is centered around basis component extraction and segment-level forecasting. As illustrated in Figure 2, TimeBase utilizes only $0.39k$ parameters, reducing MACs by $120\times$ and parameter count by more than $2.6 \times 10^3$ times compared to DLinear. Compared to the standard linear lightweight model SparseTSF, it reduces MACs by $4.5\times$, parameter count by $2.5\times$. Despite being a minimal model, TimeBase demonstrates superior predictive performance on real-world datasets of various

domains and scales. Additionally, TimeBase can also serve as a very effective plug-and-play tool for patch-based forecasting methods, enabling extreme complexity reduction, i.e., 77.74%∼93.03% for PatchTST in MACs, prompting prediction accuracy. Our contributions can be summarized as follows:

- Considering the temporal pattern similarity and low-rank characteristic, we demonstrate that basis component extraction with segment-level forecasting is an effective approach for LTSF to fully utilize both data and model. This method can significantly reduce the unavoidable ultra-high complexity and large model parameters associated with current LTSF models.

- We propose TimeBase, which is currently the **lightest time series forecaster** and an **effective plug-and-play complexity reducer**. It requires only $0.39k$ parameters, achieving $4.5\times$ reduction in MAC and a $2.5\times$ decrease in the number of parameters compared to the previously lightest model SparseTSF. Besides, it

could make **77.74%∼93.03%** computation reduction for PatchTST.

- TimeBase not only maintains an extremely small model size but also achieves competitive forecasting performance across various real-world datasets. Specifically, TimeBase ranks Top2 on 29 out of 34 average metrics (MSE and MAE) across 17 normal scale datasets when compared to the ten state-of-the-art baselines.

- TimeBase offers a potential strategy for designing LTSF models with more efficiency and could provide valuable insights for the development of backbone architectures in large pre-trained LTSF models.

## 2. Related Work

Long-term time series forecasting (LTSF) aims to predict future sequences of considerable length using extended historical windows (Ang et al., 2024). The advancement of deep learning has significantly enhanced the accuracy of LTSF, with various foundational models, such as Transformers (Zhou et al., 2021; Zhang and Yan, 2023), Temporal Convolutional Networks (TCNs) (Luo and Wang, 2024), and Recurrent Neural Networks (RNNs) (Lin et al., 2023b), being employed to design long-term forecasting networks. These models are designed based on the different properties of time series, such as series decomposition (Wu et al., 2021), frequency domain (Xu et al., 2024), and periodic characteristics (Wu et al., 2023). As to series decomposition, for instance, Autoformer (Wu et al., 2021) introduces a series decomposition block that utilizes moving average techniques to decompose complex temporal variations into seasonal and trend components, each undergoing separate time series modeling. Additionally, FEDformer (Zhou et al., 2022) further enhances the representation capabilities of the series decomposition block by employing multiple kernels moving average to decompose data at various granularities, thereby improving forecasting performance. Considering the frequency domain characteristics of time series, FITS (Xu et al., 2024) operates on the principle that time series can be manipulated through interpolation in the complex frequency domain, achieving performance comparable to state-of-the-art models for time series forecasting. On the other hand, periodicity is a significant factor considered by many LTSF methods. TimesNet (Wu et al., 2023) proposes the use of Fourier Transform to capture multiple periodic lengths of time series, expanding one-dimensional time series into several two-dimensional components, which are processed through two-dimensional networks to handle high-dimensional data. SparseTSF (Lin et al., 2024b) directly utilizes the prior periodicity, thereby reducing the scale of network parameters. CrossGNN (Huang et al., 2024b) employs moving average techniques based on periodicity to expand single-granularity time series data into

multi-granularity data, enriching the information contained within the dataset. In this paper, we propose TimeBase, which further leverages the approximate low-rank nature of long-term time series and significantly reducing the parameter scale.

## 3. Method

### 3.1. Problem Definition

In LTSF, the objective is to predict future values over an extended time horizon based on very long look-back windows. Formally, let $\mathbf{X} = [x_1, x_2, ..., x_T] \in \mathbb{R}^T$ denote the historical time series data, where $T \gg 1$ is the length of look-back window. The goal is to forecast the future values $\mathbf{Y} = [x_{T+1}, x_{T+2}, ..., x_{T+L}] \in \mathbb{R}^L$ with a forecasting horizon $L \gg 1$. However, the exceptionally long horizon scale $T$ and $L$ substantially increases model size, leading to a rapid and considerable growth in the need of computation resources, which may be unnecessary for time series data that follow simple and regular patterns. Consequently, our focus shifts to designing models that not only deliver robust and efficient performance but also remain extremely lightweight.

### 3.2. TimeBase

In practical, regular time series often exhibit prominent segment-level patterns (Lin et al., 2024b), with approximate low-rank characteristics (Jones and Brelsford, 1967). For example, traffic flow typically follows a daily period, with similar patterns recurring each day. To effectively leverage time series data and accomplish efficient forecasting, we propose **TimeBase**, implemented through **Basis Extraction** and **Segment-level Forecasting** by two extremely small-scale linear layers. This approach drastically **reduces the model parameters to the hundred level** while maintaining state-of-the-art (SOTA) forecasting performance. Most existing multivariate time series (MTS) are homogeneous, meaning that each sequence within the dataset exhibits similar patterns. Based on this property, we employ the Channel Independence (Nie et al., 2023) to simplify the forecasting of MTS into separate univariate forecasting tasks. An overview of TimeBase is shown in Figure 3.

#### 3.2.1. BASIS COMPONENT RECONSTRUCTION

First, we need to determine the basis length $P$ from the time series and segment the series accordingly. The determination process can be categorized into two scenarios: **(1) The time series has a predefined prior period much smaller than look-back window**, for instance, in domains like electricity or traffic, the cyclic patterns often follow a daily periodicity, allowing us to directly assign $P = 24$ as the segment length from hourly sampled data. **(2) When the time series lacks clear periodicity or the period exceeds**

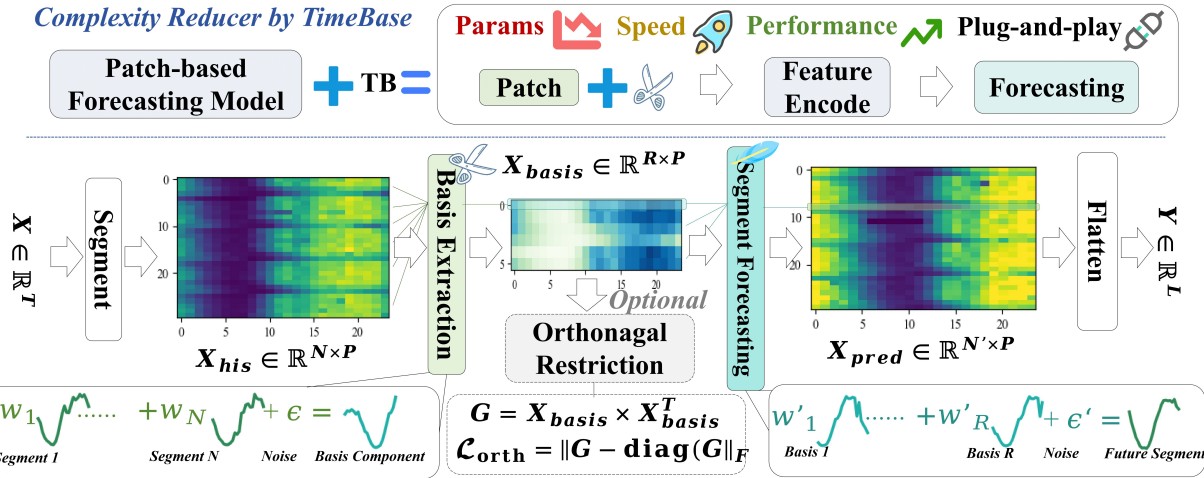

*Figure 3.* Overview of TimeBase. The core of TimeBase lies in extracting temporal basis components and segment-level forecasting. It aims to demonstrate that even the most minimalistic models can exhibit strong predictive power, providing a design foundation for more effective time series models. In addition, TimeBase can also serve as a plug-and-play reducer to decrease the complexity of any patch-based models.

**1/6 the input horizon**, frequency analysis (e.g., FFT) (Wu et al., 2023) on the training set $\mathbf{X}_{\text{train}}$ can help identify dominant components, whose wavelengths may serve as the basis length. In this case, a relatively smaller $P$ is recommended to enhance the expressiveness of basis components.

Based on the predefined basis length $P$, we divide the one-dimensional time series $\mathbf{X} \in \mathbb{R}^T$ into $N = \left\lceil \frac{T}{P} \right\rceil$ non-overlapping segments, denoted as $\mathbf{X}_{\text{his}} = [X_1, X_2, \ldots, X_N] \in \mathbb{R}^{N \times P}$, each of length $P$, analogous to non-overlapping patches (Nie et al., 2023). When the length of $X_N$ is insufficient to meet $P$, the corresponding values from $X_{N-1}$ will be used to fill in the gaps. The segment operation can be represented as:

$$\mathbf{X}_{\text{his}} = \text{Segment}_{[N,P]}(\mathbf{X}) \quad (1)$$

where $N$ and $P$ in $\text{Segment}_{[N,P]}(\cdot)$ represent the number of rows and columns of the transformed 2D matrix. The maximized rank of the matrix $\mathbf{X}_{\text{his}}$ is $R_{max} = \min(N, P)$. Given that typical time series exhibit similar temporal patterns and low-rank characteristics, we have $R \ll \min(N, P)$. In this context, directly designing a model for forecasting leads to unnecessary resource consumption. Fortunately, inspired from Basis Representation (Benson, 1998) and Linear Decomposition (Dantzig and Wolfe, 1960), a series fundamental temporal pattern can be identified, referred to as the basis components $\mathbf{X}_{\text{basis}} \in \mathbb{R}^{R \times P}$ to capture compact information and minimize the model size. Just as any vector in a coordinate system can be represented as a linear combination of its basis vectors, the combination of basis components in specific time series can represent its any segment-level temporal pattern (Hochreiter and Schmidhuber, 1997). Conversely,

we can approximate the full-rank basis components using:

$$\mathbf{X}_{\text{basis}} = \text{BasisExtract}(\mathbf{X}_{\text{his}}) \quad (2)$$

where $\text{BasisExtract}(\cdot)$ is implemented by a simple linear layer. Formally, $\mathbf{X}_{\text{his}}$ can be expressed as a linear combination of basis components, represented as $\mathbf{X}_{\text{his}}^{\top} = \mathbf{X}_{\text{basis}}^{\top} W_E + B$, where $W_E \in \mathbb{R}^{N \times R}$ is the combined weight and the bias term $B \in \mathbb{R}^R$ denote the temporal noise $\epsilon$. By rearranging, we derive $\mathbf{X}_{\text{basis}}^T = \mathbf{X}_{\text{his}}^{\top} W^{\dagger} - BW^{\dagger}$. Therefore the objective of Eq. (2) is to learn a linear layer of $W_{\text{his}} = W^{\dagger}$ and $B_{\text{his}} = -BW^{\dagger}$ to approximate the basis components. Next, we leverage the well-learned basis components to realize segment-level forecasting:

$$\mathbf{X}_{\text{pred}} = \text{SegmentForecast}(\mathbf{X}_{\text{basis}}) \quad (3)$$

Here, $\mathbf{X}_{\text{pred}} \in \mathbb{R}^{N' \times P}$ represents $N'$ future segmented time series, where $N' = \left\lceil \frac{L}{P} \right\rceil$. The operation $\text{SegmentForecast}(\cdot)$, implemented also through a linear layer, aggregates the basis components for forecasting. Finally, $\mathbf{X}_{\text{pred}}$ is unfolded to obtain prediction result $\mathbf{Y} \in \mathbb{R}^L$:

$$\mathbf{Y} = \text{Flatten}(\mathbf{X}_{\text{pred}})_{1:L} \quad (4)$$

### 3.2.2. BASIS ORTHOGONAL RESTRICTION

To make that the learned $\mathbf{X}_{\text{basis}}$ effectively captures the essential and diverse temporal patterns, an orthogonal constraint can be applied. From the perspective of the data space, the orthogonality of the basis vectors enhances its representation power, providing them ability to express as any vector in the data space through linear combination (Dantzig and Wolfe, 1960). Therefore, the temporal basis component

should also be diverse and distinct, preventing the extraction of very single time-series patterns. Based on this, we apply the Basis Orthogonal Restriction.

Specifically, we penalize the deviation of $\mathbf{X}_{\text{basis}}$ from an orthogonal set by adding a regularization loss $\mathcal{L}_{\text{orth}}$:

$$\mathbf{G} = \mathbf{X}_{\text{basis}}^{\top}\mathbf{X}_{\text{basis}} \tag{5}$$

$$\mathcal{L}_{\text{orth}} = \|\mathbf{G} - \text{diag}(\mathbf{G})\|_F^2 \tag{6}$$

where $\mathbf{G}$ is the gram matrix of $\mathbf{X}_{\text{basis}}$ and $\|\cdot\|_F$ denotes the Frobenius norm. This term encourages $\mathbf{X}_{\text{basis}}$ to approach an orthogonal configuration, ensuring that each basis component captures unique and uncorrelated temporal patterns. The overall training objective is then updated to:

$$\mathcal{L} = \mathcal{L}_{\text{prediction}} + \lambda_{\text{orth}}\mathcal{L}_{\text{orth}} \tag{7}$$

Here, $\mathcal{L}_{\text{prediction}}$ represents the original prediction loss, i.e., mean squared error (MSE) for regression, and $\lambda_{\text{orth}}$ is a hyperparameter that controls the weight of the orthogonal regularization term.

### 3.3. Parameter Scale of TimeBase

**Theorem 3.1** (Parameter Scale of TimeBase). *Let $T$ denote the length of look-back window, $L$ is the length of the forecast, $P$ represents the length of the segment, and $R$ gives the number of basis components. The parameter scale of TimeBase can be expressed as:*

$$Number = \underbrace{\frac{R}{P}}_{a}\times T + \underbrace{\frac{R+1}{P}}_{b}\times L + R \tag{8}$$

Theorem 3.1 shows that the parameter scale of *TimeBase* grows linearly with both the look-back window length $T$ and the forecast horizon $L$. In a typical long-term forecasting setup where $T = L = 720$, the number of parameters in TimeBase, with a complexity of $\mathcal{O}(aT+bL)$, is significantly smaller than that of DLinear (Zeng et al., 2023), which requires $2TL$ parameters, and SparseTSF (Lin et al., 2024b), which uses $\frac{TL}{P^2} + P$, both with $\mathcal{O}(TL)$ complexity.

## 4. Experiment

In this section, we demonstrate the advantages of Time-Base in **competitive forecasting performance, extremely light efficiency and very effective plug-and-play function.** More experiment details and additional experiment results are available at Appendix C.

### 4.1. Experiment Setup

**Datasets** We conduct experiments on **21 widely-used and publicly available real-world datasets**, including **17 normal-scale benchmarks**: ETTh1, ETTh2, ETTm1, ETTm2[1], Weather[2], Electricity[3], Traffic[4],Solar Energy (Lai et al., 2018), Wind (Li et al., 2022), , METR-LA (Li et al., 2017), Exchange Rate (Lai et al., 2018), ZafNoo (Poyatos et al., 2020) and CzeLan (Poyatos et al., 2020), AQShunyi (Zhang et al., 2017), AQWan (Zhang et al., 2017), and **4 very large datasets**: CA (4.52B), GLA (2.02B), GBA (1.24B),SD (0.38B) (Liu et al., 2024c). Adhering to the established protocol in (Wu et al., 2021; Qiu et al., 2024; Liu et al., 2024c), we partition the datasets into training, validation, and test sets with a ratio of 6:2:2 for four ETT datasets, CA, GLA, GBA, SD, and 7:1:2 for the remaining datasets. The statics of dataset is summarized in Table 1.

*Table 1.* Dataset Statistics. Var is the number of variables, Length is the dataset length, $T$ is the length of look-back window, $L$ is the forecasting horizon, Freq is the sampling frequency, Scale represents the number of data points.

| | Dataset | Var | Length | $T$ | $L$ | Freq | Scale |
|---|---|---|---|---|---|---|---|
| Normal Scale Benchmark | ETTh1 | 7 | 14,400 | 720 | 96∼720 | 1hour | 0.1M |
| | ETTh2 | 7 | 14,400 | 720 | 96∼720 | 1hour | 0.1M |
| | ETTm1 | 7 | 57,600 | 720 | 96∼720 | 15mins | 0.4M |
| | ETTm2 | 7 | 57,600 | 720 | 96∼720 | 15mins | 0.4M |
| | Weather | 21 | 52,696 | 720 | 96∼720 | 10mins | 1.1M |
| | Electricity | 321 | 26,304 | 720 | 96∼720 | 1hour | 8.1M |
| | Traffic | 862 | 17,544 | 720 | 96∼720 | 1hour | 15.0M |
| | Solar | 137 | 52,560 | 720 | 96∼720 | 10mins | 7.2M |
| | Wind | 7 | 48,673 | 720 | 96∼720 | 15mins | 0.4M |
| | METR-LA | 207 | 34,272 | 720 | 96∼720 | 5mins | 7.1M |
| | Exchange | 8 | 7,588 | 720 | 96∼720 | 1day | 60.7K |
| | AQshunyi | 11 | 35,064 | 720 | 96∼720 | 1hour | 0.4M |
| | AQWan | 11 | 35,064 | 720 | 96∼720 | 1hour | 0.4M |
| | ZafNoo | 11 | 19,225 | 720 | 96∼720 | 30mins | 0.2M |
| | CzeLan | 11 | 19,934 | 720 | 96∼720 | 30mins | 0.2M |
| | PM2.5 | 184 | 11,688 | 720 | 96∼720 | 3hours | 2.2M |
| | Temp | 184 | 11,688 | 720 | 96∼720 | 3hours | 2.2M |
| Large | CA | 8600 | 525,888 | 720 | 96∼720 | 5mins | 4820G |
| | GLA | 3834 | 525,888 | 720 | 96∼720 | 5mins | 2020G |
| | GBA | 2352 | 525,888 | 720 | 96∼720 | 5mins | 1240G |
| | SD | 716 | 525,888 | 720 | 96∼720 | 5mins | 380G |

**Baselines** We compare TimeBase with 10 baselines, which comprise the SOTA long-term forecasting models: TimeMixer (Wang et al., 2024d), iTransformer (Liu et al., 2024a), PatchTST (Nie et al., 2023), DLinear (Zeng et al., 2023), TimesNet (Wu et al., 2023), FEDformer (Zhou et al., 2022), Autoformer (Wu et al., 2021), and Informer (Zhou et al., 2021), relatively efficient models: FITS (Xu et al., 2024) and SparseTSF (Lin et al., 2024b).

---

[1] https://github.com/zhouhaoyi/ETDataset
[2] https://www.bgc-jena.mpg.de/wetter
[3] https://archive.ics.uci.edu/ml/datasets
[4] https://pems.dot.ca.gov/

*Table 2.* Long-term time series forecasting results in 17 normal scale datasets, comparing TimeBase with other baselines. Results are averaged across different forecasting horizons $L \in \{96, 192, 336, 720\}$. The input length $H$ is set to 720 across all models. Best results are marked in red ; second-best results are underlined in blue .

| Methods | TimeBase (ours) | | SparseTSF (2024b) | | TimeMixer (2024d) | | FITS (2024) | | iTransformer (2024a) | | DLinear (2023) | | PatchTST (2023) | | TimesNet (2023) | | FEDformer (2022) | | Autoformer (2021) | | Informer (2021) | |
|---|---|---|---|---|---|---|---|---|---|---|---|---|---|---|---|---|---|---|---|---|---|---|
| Metric | MSE | MAE | MSE | MAE | MSE | MAE | MSE | MAE | MSE | MAE | MSE | MAE | MSE | MAE | MSE | MAE | MSE | MAE | MSE | MAE | MSE | MAE |
| ETTh1 | 0.396 | 0.414 | 0.407 | 0.419 | 0.454 | 0.474 | 0.417 | 0.430 | 0.454 | 0.467 | 0.437 | 0.448 | 0.420 | 0.440 | 0.505 | 0.499 | 0.523 | 0.524 | 0.726 | 0.641 | 1.445 | 0.930 |
| ETTh2 | 0.347 | 0.397 | 0.344 | 0.386 | 0.379 | 0.426 | 0.334 | 0.382 | 0.392 | 0.422 | 0.479 | 0.471 | 0.344 | 0.391 | 0.433 | 0.455 | 0.429 | 0.470 | 1.086 | 0.802 | 5.486 | 1.892 |
| ETTm1 | 0.356 | 0.380 | 0.362 | 0.384 | 0.381 | 0.414 | 0.359 | 0.382 | 0.370 | 0.401 | 0.361 | 0.383 | 0.355 | 0.385 | 0.408 | 0.425 | 0.438 | 0.466 | 0.564 | 0.522 | 1.138 | 0.818 |
| ETTm2 | 0.250 | 0.314 | 0.252 | 0.316 | 0.283 | 0.347 | 0.252 | 0.314 | 0.278 | 0.338 | 0.271 | 0.337 | 0.251 | 0.319 | 0.300 | 0.354 | 0.409 | 0.462 | 0.431 | 0.463 | 3.594 | 1.473 |
| Weather | 0.219 | 0.263 | 0.244 | 0.286 | 0.238 | 0.281 | 0.244 | 0.286 | 0.233 | 0.273 | 0.242 | 0.299 | 0.223 | 0.264 | 0.254 | 0.293 | 0.355 | 0.391 | 0.446 | 0.457 | 0.567 | 0.513 |
| Electricity | 0.167 | 0.258 | 0.168 | 0.264 | 0.171 | 0.274 | 0.172 | 0.266 | 0.166 | 0.264 | 0.169 | 0.272 | 0.169 | 0.266 | 0.238 | 0.334 | 0.235 | 0.348 | 0.236 | 0.343 | 0.408 | 0.464 |
| Traffic | 0.418 | 0.278 | 0.414 | 0.280 | 0.419 | 0.300 | 0.426 | 0.291 | 0.406 | 0.290 | 0.418 | 0.285 | 0.394 | 0.266 | 0.641 | 0.346 | 0.638 | 0.400 | 0.684 | 0.421 | 1.028 | 0.588 |
| AQShunyi | 0.674 | 0.506 | 0.760 | 0.546 | 0.736 | 0.536 | 0.763 | 0.548 | 0.722 | 0.520 | 0.757 | 0.572 | 0.690 | 0.503 | 0.788 | 0.559 | 1.130 | 0.757 | 1.392 | 0.878 | 1.688 | 0.966 |
| AQWan | 0.779 | 0.499 | 0.826 | 0.526 | 0.822 | 0.521 | 0.813 | 0.519 | 0.843 | 0.538 | 0.883 | 0.560 | 0.810 | 0.513 | 0.857 | 0.543 | 0.932 | 0.565 | 1.001 | 0.654 | 1.123 | 0.744 |
| CzeLan | 0.225 | 0.262 | 0.240 | 0.292 | 0.234 | 0.282 | 0.250 | 0.304 | 0.245 | 0.296 | 0.242 | 0.290 | 0.229 | 0.275 | 0.258 | 0.312 | 0.266 | 0.314 | 0.289 | 0.368 | 0.347 | 0.426 |
| ZafNoo | 0.495 | 0.445 | 0.531 | 0.490 | 0.505 | 0.464 | 0.520 | 0.480 | 0.540 | 0.500 | 0.540 | 0.495 | 0.510 | 0.469 | 0.545 | 0.502 | 0.601 | 0.539 | 0.667 | 0.635 | 0.726 | 0.696 |
| Exchange | 0.309 | 0.392 | 0.316 | 0.410 | 0.330 | 0.426 | 0.302 | 0.400 | 0.301 | 0.401 | 0.312 | 0.410 | 0.308 | 0.400 | 0.326 | 0.425 | 0.380 | 0.472 | 0.447 | 0.589 | 0.432 | 0.581 |
| METR-LA | 1.253 | 0.754 | 1.302 | 0.764 | 1.270 | 0.740 | 1.296 | 0.760 | 1.253 | 0.734 | 1.300 | 0.759 | 1.210 | 0.706 | 1.390 | 0.812 | 1.550 | 0.876 | 1.639 | 0.974 | 1.894 | 1.146 |
| PM2.5 | 0.428 | 0.437 | 0.467 | 0.477 | 0.465 | 0.474 | 0.460 | 0.469 | 0.480 | 0.492 | 0.449 | 0.457 | 0.458 | 0.466 | 0.466 | 0.475 | 0.560 | 0.547 | 0.621 | 0.650 | 0.636 | 0.663 |
| Solar | 0.216 | 0.254 | 0.216 | 0.264 | 0.244 | 0.296 | 0.229 | 0.279 | 0.233 | 0.285 | 0.227 | 0.276 | 0.226 | 0.275 | 0.243 | 0.295 | 0.252 | 0.291 | 0.306 | 0.380 | 0.327 | 0.411 |
| Temp | 0.187 | 0.338 | 0.295 | 0.412 | 0.302 | 0.423 | 0.315 | 0.442 | 0.302 | 0.423 | 0.292 | 0.418 | 0.291 | 0.406 | 0.304 | 0.427 | 0.343 | 0.456 | 0.404 | 0.587 | 0.471 | 0.682 |
| Wind | 0.940 | 0.709 | 1.014 | 0.736 | 1.044 | 0.758 | 1.023 | 0.745 | 1.025 | 0.748 | 1.018 | 0.739 | 0.968 | 0.703 | 1.101 | 0.799 | 1.230 | 0.841 | 1.237 | 0.947 | 1.381 | 1.033 |

## 4.2. Forecsating Performance of TimeBase

As shown in Table 2, across 17 normal-scale benchmarks, TimeBase achieves Top-2 performance on 16 datasets, with the only exception being ETTh2, where its accuracy still remains highly competitive. These results further demonstrate the robustness and effectiveness of TimeBase across a wide range of forecasting tasks. Compared to the standard linear model DLinear, TimeBase reduces MSE and MAE by 8.82% and 7.64%, respectively. When compared with the currently most lightweight models, FITS and SparseTSF, TimeBase achieves an average reduction of 6.15∼6.34% in MSE and 4.85∼5.44% in MAE. Furthermore, against some of the most powerful models to date, such as TimeMixer, iTransformer, and PatchTST, TimeBase achieves the best results on 23 out of 34 average forecasting metrics and ranks second on 6, when carefully tuned. Despite its extremely compact size, TimeBase delivers competitive forecasting accuracy, thanks to its compact yet effective modeling design.

## 4.3. Efficiency Analysis

**Main Efficiency Comparision**   In addition to its impressive predictive performance, TimeBase offers another major advantage: its exceptionally lightweight design. Here, we provide a more comprehensive comparison, examining both static and runtime metrics, which include **(1) Parameters:** *The total number of trainable parameters, reflecting the model's size.* **(2) MACs (Multiply-Accumulate Operations):** *A standard measure of computational complexity in neural networks, representing the number of multiply-accumulate operations required by the model.* **(3) Max Memory:** *The peak memory usage during training.* **(4) Epoch Time:** *The time required to train the model for one epoch, averaged over three runs.* **(5) Infer Time:** *Infer Time indicates the average inference time per sample on CPU.*

The look-back window for each model are set as 720, and the max memory is recorded with a constant batch size of 12. The efficiency analysis presented in Table 3 highlights the remarkable advantages of TimeBase over other state-of-the-art models in terms of both static and runtime metrics. TimeBase achieves a substantial reduction in the number of parameters and computational complexity compared to many parameter-heavy models. Specifically, compared to lightest model SparseTSF, TimeBase reduces the parameter count by up to 61% and the MACs by over 78%, while also using significantly less memory (29% reduction) and training faster (34% reduction in epoch time). These

*Table 3.* Efficiency comparison of TimeBase and other state-of-the-art models on the Electricity dataset with a forecasting length of 720. To ensure fair comparison, the look-back window is set as 720 for all models. MACs here indicate the computational cost for processing a single sample, and memory usage is measured with a batch size of 12 for all models.

| Model | Parameters | MACs | Max Mem.(MB) | Epoch Time(s) | Infer Time (CPU) |
|---|---|---|---|---|---|
| Informer (2021) | 22.45M | 7.85G | 1424.99 | 143.05 | 72.67ms |
| Autoformer (2021) | 22.14M | 8.97G | 4348.89 | 225.78 | 126.75ms |
| FEDformer (2022) | 22.14M | 10.48G | 2361.76 | 558.03 | 203.31ms |
| PatchTST (2023) | 8.69M | 14.17G | 18034.33 | 827.34 | 249.02ms |
| DLinear (2023) | 1.04M | 333.04M | 158.21 | 41.08 | 3.25ms |
| FITS (2024) | 133.60K | 42.73M | 496.70 | 43.13 | 3.95ms |
| iTransformer (2024a) | 5.47M | 1.79G | 828.32 | 65.62 | 30.41ms |
| TimeMixer (2024d) | 5.58M | 30.30G | 1376.14 | 537.26 | 71.59ms |
| SparseTSF (2024b) | 1.00K | 12.71M | 125.20 | 31.30 | 2.59ms |
| TimeBase(ours) | **0.39K** | **2.77M** | **88.89** | **20.60** | **0.98ms** |
| *Reduction* | *2.56×↓* | *4.58×↓* | *1.40×↓* | *1.51×↓* | *2.64×↓* |

results further demonstrate that TimeBase not only maintains strong predictive performance but also offers superior efficiency, making it well-suited for resource-constrained environments.

**Efficiency in Ultra-long Look-back Window**    Additionally, we evaluate the efficiency of TimeBase under ultra-long look-back windows, comparing it with DLinear. The comparison focuses on three key metrics: running time per iteration, GPU memory usage, and parameter count, as shown in Figure 4. As the look-back window increases from 720 to 6480, with a fixed batch size of 12 and prediction length of 720, TimeBase consistently demonstrates its lightweight nature. Even with a ninefold increase in input sequence length, TimeBase's running time only increases by 0.05 seconds, GPU memory usage expands by a factor of 3.8, and the number of parameters grows by only 3.1 times. These results highlight the model's extreme efficiency and scalability in handling ultra-long sequences.

### 4.4. Performance on Large-Scale Datasets

To further validate the robustness of TimeBase under extreme conditions, we conduct experiments on four large-scale datasets. As shown in Table 5, TimeBase consistently outperforms or matches strong baselines across all forecasting horizons, underscoring its effectiveness in handling large-scale forecasting tasks. Notably, TimeBase ranks within the top two across all horizons, highlighting its advantages in both prediction quality and computational efficiency. Compared to state-of-the-art models such as iTransformer, DLinear, and PatchTST, TimeBase achieves lower forecasting errors while significantly reducing computational cost, as reflected by its much lower Multiply–Accumulate Operations (MACs). Even on the largest dataset, CA with 8600 variables, TimeBase requires only 42.11M∼74.30M MACs, showcasing its exceptional capability in scenarios

with limited computational resources or extremely large input scales.

### 4.5. Plug-and-Play Complexity Reducer for PatchTST

**Settings**    TimeBase can be seamlessly integrated into any patch-based forecasting model to extract basis components prior to patching the input time series. For patch-based methods, the general prediction pipeline can be abstracted as $\mathbf{Y} = \text{Head}(\text{Encode}(\text{Patch}(\mathbf{X})))$. With the incorporation of TimeBase, the pipeline becomes $\mathbf{Y} = \text{Head}(\text{Encode}(\text{BasisExtract}(\text{Patch}(\mathbf{X}))))$, where the extraction of basis components significantly reduces the number of required patches, thus greatly simplifying model complexity. We validate the plug-and-play capability of TimeBase by applying it to the widely recognized PatchTST model, resulting in the variant PATCHTST (W/ TIMEBASE). To ensure a fair comparison in terms of forecasting accuracy, computational complexity, and parameter count, we set the input sequence length to 720 for both PatchTST and PATCHTST (W/ TIMEBASE).

**Quantitative Analysis of TimeBase Integration**    As shown in Table 4, PATCHTST (W/ TIMEBASE) achieves comparable or even slightly improved forecasting accuracy, measured by MSE and MAE, while significantly reducing both parameter count and computational cost. Specifically, across 56 forecasting metrics on 7 datasets, PATCHTST (W/ TIMEBASE) outperforms the original PatchTST on 43 metrics, while the remaining 13 show only a marginal average degradation of 1.54%, which can be considered negligible. In terms of overall performance, PATCHTST (W/ TIMEBASE) reduces MSE and MAE by **1.27%** and **1.13%**, respectively, indicating that the basis extraction does not harm predictive power, and may even offer slight improvements. On the efficiency front, TimeBase enables dramatic reductions: MACs are reduced by **77.74%** to **93.03%**, and the

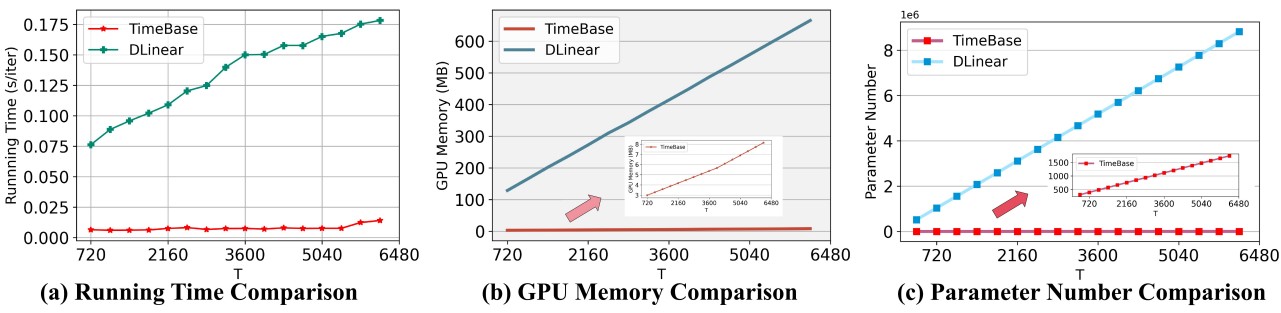

(a) Running Time Comparison  (b) GPU Memory Comparison  (c) Parameter Number Comparison

*Figure 4.* Comparison of efficiency metrics between TimeBase and other lightweight model with varying look-back windows. (a) Running time per iteration (s/iter), (b) GPU memory consumption, and (c) Parameter count as the look-back window increases from 720 to 6480.

*Table 4.* Performance of TimeBase as a Plug-and-Play Component for Patch-Based Methods. The input length is set as 720.

| Model | | PatchTST | | | PATCHTST (W/ TIMEBASE) | | | |
|---|---|---|---|---|---|---|---|---|
| Metric | | MSE | MAE | MACs | Params | MSE | MAE | MACs | Params |

| | | MSE | MAE | MACs | Params | MSE | MAE | MACs | Params |
|---|---|---|---|---|---|---|---|---|---|
| ETTh1 | 96 | 0.377 | 0.408 | 11.05 M | 0.15 M | **0.364**↓3.45% | **0.398**↓2.45% | **0.77M**↓93.03% | **0.03M**↓80.00% |
| | 192 | 0.413 | 0.431 | 12.02 M | 0.29 M | **0.402**↓2.66% | **0.424**↓1.62% | **0.84M**↓93.01% | **0.04M**↓86.21% |
| | 336 | 0.436 | 0.446 | 13.47 M | 0.50 M | **0.423**↓2.98% | **0.437**↓2.02% | **1.24M**↓90.79% | **0.06M**↓88.00% |
| | 720 | **0.455** | **0.475** | 17.34 M | 1.05 M | 0.475↑4.40% | 0.49↑3.16% | **1.58M**↓90.89% | **0.11M**↓89.52% |
| ETTh2 | 96 | 0.276 | **0.339** | 11.05 M | 0.15 M | **0.275**↓0.36% | 0.339↓0.00% | **1.28M**↓88.42% | **0.03M**↓80.00% |
| | 192 | 0.342 | 0.385 | 12.02 M | 0.29 M | **0.334**↓2.34% | **0.381**↓1.04% | **1.39M**↓88.44% | **0.05M**↓82.76% |
| | 336 | 0.364 | **0.405** | 13.47 M | 0.50 M | **0.36**↓1.10% | 0.407↑0.49% | **1.25M**↓90.72% | **0.06M**↓88.00% |
| | 720 | **0.395** | **0.434** | 17.34 M | 1.05 M | 0.397↑0.51% | 0.436↑0.46% | **1.19M**↓93.14% | **0.09M**↓91.43% |
| ETTm1 | 96 | 0.298 | 0.352 | 258.69 M | 1.51 M | **0.29**↓2.68% | **0.345**↓1.99% | **28.87M**↓88.84% | **0.52M**↓65.56% |
| | 192 | 0.335 | 0.373 | 266.43 M | 2.61 M | **0.331**↓1.19% | **0.368**↓1.34% | **29.73M**↓88.84% | **0.65M**↓75.10% |
| | 336 | 0.366 | 0.394 | 278.05 M | 4.27 M | **0.364**↓0.55% | **0.386**↓2.03% | **31.02M**↓88.84% | **0.83M**↓80.56% |
| | 720 | 0.420 | 0.421 | 309.01 M | 8.69 M | **0.419**↓0.24% | **0.416**↓1.19% | **34.46M**↓88.85% | **1.32M**↓84.81% |
| ETTm2 | 96 | **0.165** | 0.260 | 258.69 M | 1.51 M | 0.165↓0.00% | **0.256**↓1.54% | **57.59M**↓77.74% | **0.65M**↓56.95% |
| | 192 | **0.219** | 0.298 | 266.43 M | 2.61 M | 0.222↑1.37% | **0.293**↓1.68% | **29.73M**↓88.84% | **0.65M**↓75.10% |
| | 336 | **0.268** | 0.333 | 278.05 M | 4.27 M | 0.273↑1.87% | **0.332**↓0.30% | **61.89M**↓77.74% | **1.26M**↓70.49% |
| | 720 | **0.352** | 0.386 | 309.01 M | 8.69 M | 0.353↑0.28% | **0.385**↓0.26% | **68.77M**↓77.75% | **2.24M**↓74.22% |
| Weather | 96 | 0.149 | 0.199 | 776.08 M | 1.51 M | **0.145**↓2.68% | **0.195**↓2.01% | **86.60M**↓88.84% | **0.52M**↓65.56% |
| | 192 | 0.193 | 0.243 | 799.30 M | 2.61 M | **0.189**↓2.07% | **0.238**↓2.06% | **89.18M**↓88.84% | **0.65M**↓75.10% |
| | 336 | **0.240** | **0.281** | 834.14 M | 4.27 M | 0.243↑1.25% | 0.284↑1.07% | **92.62M**↓88.90% | **0.83M**↓80.56% |
| | 720 | **0.312** | **0.334** | 927.04 M | 8.69 M | 0.314↑0.64% | 0.334↓0.00% | **103.38M**↓88.85% | **1.32M**↓84.81% |
| Electricity | 96 | 0.141 | 0.240 | 11.86 G | 1.51 M | **0.128**↓9.22% | **0.223**↓7.08% | **1.32G**↓88.87% | **0.52M**↓65.56% |
| | 192 | 0.156 | 0.256 | 12.22 G | 2.61 M | **0.145**↓7.05% | **0.238**↓7.03% | **1.36G**↓88.87% | **0.65M**↓75.10% |
| | 336 | 0.172 | 0.267 | 12.75 G | 4.27 M | **0.160**↓6.98% | **0.255**↓4.49% | **1.42G**↓88.86% | **0.83M**↓80.56% |
| | 720 | 0.207 | 0.299 | 14.17 G | 8.69 M | **0.197**↓4.83% | **0.288**↓3.68% | **1.58G**↓88.85% | **1.32M**↓84.81% |
| Traffic | 96 | 0.363 | **0.250** | 31.86 G | 1.51 M | **0.360**↓0.83% | 0.252↑0.80% | **4.27G**↓86.60% | **0.55M**↓63.58% |
| | 192 | 0.382 | 0.258 | 32.81 G | 2.61 M | **0.371**↓2.88% | **0.256**↓0.78% | **4.39G**↓86.62% | **0.70M**↓73.18% |
| | 336 | 0.399 | **0.268** | 34.24 G | 4.27 M | **0.396**↓0.75% | 0.278↑3.73% | **4.58G**↓86.62% | **0.92M**↓78.45% |
| | 720 | 0.432 | 0.289 | 38.05 G | 8.69 M | **0.422**↓2.31% | **0.284**↓1.73% | **5.09G**↓86.62% | **1.51M**↓82.62% |

number of parameters is decreased by **56.95%** to **91.43%**. For example, on the Traffic dataset, the parameter count of PatchTST ranges from 1.51M to 8.69M, whereas the TimeBase-enhanced variant achieves the same task with substantially fewer parameters and computations. These results demonstrate that the low-rank nature of long-term time series often leads to substantial computational redundancy, which can be effectively mitigated by TimeBase via basis component extraction. Its plug-and-play design al-

lows seamless integration with existing patch-based models, enabling users to boost efficiency significantly without the need for architectural re-design or retraining from scratch.

### 4.6. Hyperparameter Analysis

This section explores the impact of two key hyperparameters on the performance of TimeBase: the number of basis functions $R$ and the orthogonal loss weight $\lambda_{orth}$. Fig-

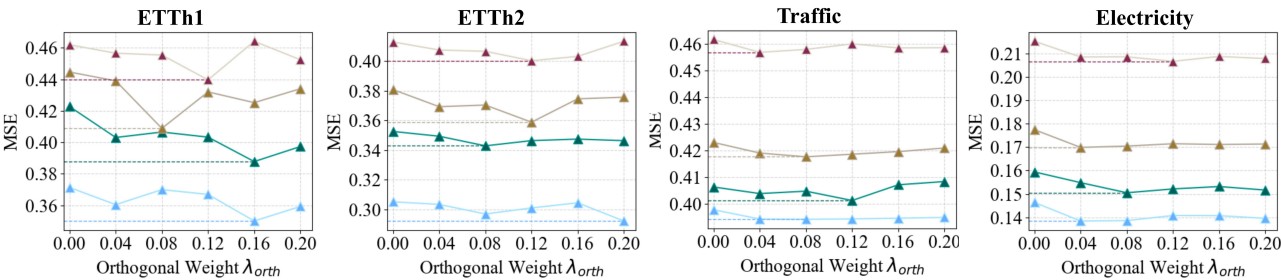

*Figure 5.* Effect of orthogonal loss weight $\lambda_{orth}$ across Traffic, Electricity, ETTh1, and ETTh2.

*Table 5.* Long-term time series forecasting results in large scale datasets. Input lengths of all models are set to 720.

| Methods | | TimeBase | | iTransformer | | DLinear | | PatchTST | |
|---|---|---|---|---|---|---|---|---|---|
| Metric | | MAE | MACs | MAE | MACs | MAE | MACs | MAE | MACs |
| CA | 96 | **0.259** | 42.11M | 0.309 | 7.24G | 0.279 | 1.78G | **0.266** | 7.61G |
| | 192 | **0.299** | 47.06M | 0.340 | 7.45G | **0.312** | 3.57G | 0.358 | 15.22G |
| | 336 | **0.317** | 54.49M | 0.337 | 7.76G | 0.355 | 6.24G | **0.321** | 26.63G |
| | 720 | **0.386** | 74.30M | 0.441 | 8.61G | **0.386** | 13.38G | 0.439 | 57.07G |
| GLA | 96 | **0.302** | 18.77M | 0.361 | 3.23G | 0.358 | 795.39M | **0.314** | 3.39G |
| | 192 | **0.338** | 20.98M | 0.345 | 3.32G | **0.385** | 1.59G | 0.397 | 6.78G |
| | 336 | **0.397** | 24.29M | 0.413 | 3.46G | 0.447 | 2.78G | **0.407** | 11.87G |
| | 720 | **0.452** | 33.13M | 0.489 | 3.84G | 0.488 | 5.97G | 0.473 | 25.44G |
| GBA | 96 | **0.183** | 11.52M | 0.206 | 1.98G | **0.189** | 487.94M | 0.198 | 2.08G |
| | 192 | **0.194** | 12.87M | 0.198 | 2.04G | **0.212** | 975.87M | 0.224 | 4.16G |
| | 336 | **0.199** | 14.90M | 0.231 | 2.13G | 0.237 | 1.71G | **0.208** | 7.28G |
| | 720 | **0.206** | 20.32M | 0.229 | 2.36G | **0.237** | 3.66G | 0.227 | 15.61G |
| SD | 96 | **0.163** | 3.51M | 0.194 | 606.99M | 0.187 | 148.54M | **0.175** | 633.47M |
| | 192 | **0.173** | 3.92M | 0.199 | 624.73M | **0.184** | 297.08M | 0.174 | 1.27G |
| | 336 | **0.179** | 4.54M | **0.185** | 651.35M | 0.208 | 519.88M | 0.195 | 2.22G |
| | 720 | **0.183** | 6.19M | 0.216 | 722.32M | **0.202** | 1.11G | 0.218 | 4.75G |

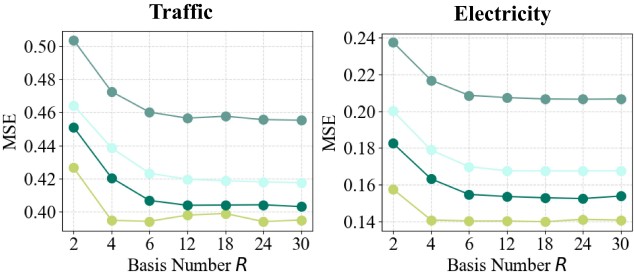

*Figure 6.* Effect of basis number $R$.

ure 5 shows the MSE results as the orthogonal loss weight $\lambda_{orth}$ is varied across [0.00, 0.04, 0.08, 0.12, 0.16, 0.20]. For datasets such as ETTh1, and ETTh2, prediction performance fluctuates with different values of $\lambda_{orth}$. However, for Traffic and Electricity, the performance remains relatively stable. Overall, an appropriate value of $\lambda_{orth}$ can enhance forecasting performance, which are varied across different datasets. In Figure 6, the number of basis number $R$ is varied from [2, 4, 6, 12, 18, 24, 30], and the corresponding MSE results for Traffic, Electricity datasets are reported. The results show that using too few basis components (e.g.,

$R = 2$) leads to a noticeable performance drop. However, once $R$ exceeds a modest threshold (typically $R \geq 4$), the model performance stabilizes. These findings indicate that TimeBase can achieve strong forecasting performance with a small number of basis components. In practice, we typically set $R = 6$ for most datasets. For datasets with more complex temporal patterns, a larger $R$ is recommended to enhance the expressive capacity of the basis components.

## 5. Conclusion

Considering the temporal pattern similarity and low-rank characteristics, we design TimeBase, the lightest known model for long-term forecasting, with only 0.39K parameters, 2.77M MACs computation, 88.89M memory usage, and a CPU inference speed of 0.98ms. This demonstrates that even the most minimalist models can achieve strong predictive performance, providing a design foundation for more efficient time series models. Furthermore, TimeBase can function as a plug-and-play tool to reduce the complexity of any patch-based models. This approach (1) enables long-term time series forecasting models to be deployed on resource-constrained edge devices, and (2) offers new insights for lightweight model designs, encouraging time series researchers to fully leverage sequential data and inspiring the development of backbone networks for pre-trained large LTSF models.

## Acknowledgement

This work was supported by the National Natural Science Foundation of China (No.12227901), Natural Science Foundation of Jiangsu Province (BK.20240460, BK.20240461), the grant from State Key Laboratory of Resources and Environmental Information System. The AI-driven experiments, simulations and model training are performed on the robotic AI-Scientist platform of Chinese Academy of Science.

## Impact Statement

This paper aims to advance the field of machine learning. The research may have various social impacts, but we believe it is not necessary to emphasize them here.

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

# A. Lightweight Forecasting Survey

As for lightweight forecasting , many Linear-based models have emerged, aiming to achieve lightweight forecasting solutions (Wang et al., 2024a). The main differences of them from TimeBase are summarized in Table 6. DLinear (Zeng et al., 2023) introduces a linear model based on trend and seasonal decomposition, whose competitive forecasting performance empirically demonstrated the feasibility of using Linear for LTSF. Following this, TiDE (Das et al., 2024) provides theoretical proof that the simplest linear analogue could achieve near-optimal error rates for linear dynamical systems. Later, numerous Mixer-based works emerge, such as MTS-Mixer (Li et al., 2023), TSMixer (Ekambaram et al., 2023) and HDMixer (Huang et al., 2024a), which stack standard MLP layers to efficiently capture correlations across different dimensions of multivariate time series. Furthermore, Koopa (Liu et al., 2024b) addresses the challenge of dynamic and unstable time series systems by disentangling time-variant and time-invariant components using Fourier filters and designing a Koopman Predictor to advance the respective dynamics. TimeMixer (Wang et al., 2024d) tackles the issue of different granularity levels in micro and macro series by proposing mixing blocks, fully leveraging disentangled multi-scale series in both past extraction and future prediction phases. These works represent efficient time series forecasting models based on Linear structures (1.03 M $\sim$ 31.07 M). More recently, FITS (Xu et al., 2024) further reduces model complexity to 10K parameters by employing low-pass filtering to extract a compact set of frequency-domain features via linear layers, while SparseTSF (Lin et al., 2024b) leverages periodic structures in the data to achieve an even more compact design with only 1.0K parameters. However, it still remains challenging when faced with stricter deployment constraints on edge devices and higher efficiency demands (Chatfield, 2013). To address this, we propose TimeBase to significantly reduce data complexity by extracting temporal basis components using the low-rank characteristics in long-term time series. Our approach could require only $0.39K$ parameters to achieve competitive predictive performance.

*Table 6.* Differences between TimeBase and other lightweight linear-based models. Model size refers to the parameter count, and performance is evaluated using the MSE metric, both assessed in the 720-horizon electricity forecasting task.

| Linear-based Model | TimeBase(Ours) | SparseTSF | FITS | TimeMixer | Koopa | DLinear | MTS-Mixer | TSMixer | HDMixer | TiDE |
|---|---|---|---|---|---|---|---|---|---|---|
| Model Size | **Extremely Light** (0.39 K) | Extremely Light (1.0 K) | Extremely Light (10 K) | Light (5.57M) | Normal (30.04 M) | Light (1.03M) | Light (2.02M) | Light (1.05M) | Light (4.81M) | Normal (31.07M) |
| Performance | **Perfect** (0.208) | Perfect (0.208) | Perfect (0.209) | Perfect (0.209) | Perfect (0.215) | Perfect (0.209) | Good (0.213) | Good (0.236) | Good (0.243) | Good (0.241) |
| Forecasting Type | **Segment-level** | Segment-level | Frequency-level | Point-level | Point-level | Point-level | Point-level | Point-level | Point-level | Point-level |
| Plug-and-play | ✓ | ✗ | ✗ | ✗ | ✗ | ✗ | ✗ | ✗ | ✗ | ✗ |

# B. Effectiveness Analysis of TimeBase

**Low Rank Characteristics**  Assume we have a historical time series matrix $\mathbf{X}_{\text{his}} \in \mathbb{R}^{N \times P}$, where $N$ represents the number of temporal segments and $P$ denotes the length of each segment. According to the definition of singular value decomposition (SVD), any matrix $\mathbf{X} \in \mathbb{R}^{N \times P}$ can be decomposed as:

$$\mathbf{X}_{\text{his}} = U\Sigma V^T \tag{9}$$

where $U$ is an $N \times N$ orthogonal matrix, $\Sigma$ is an $N \times P$ diagonal matrix containing the singular values, and $V$ is an $P \times P$ orthogonal matrix. We can establish a threshold to compute the approximate rank:

$$\text{Rank}(\mathbf{X}_{\text{his}}) \approx \#\{\sigma_i \in \Sigma : \sigma_i > \epsilon\} \tag{10}$$

where $\sigma_i$ is the singular values of $\mathbf{X}_{\text{his}}$ for a small threshold $\epsilon > 0$, and $\#\{\cdot\}$ is the number. Figure 1 (b) illustrates the singular value distribution from real-world long-term time series data. The rapid decay of the singular values indicates that a large portion of the matrix's information is captured by only a few dominant components, thereby confirming its low-rank nature. Thus, Due to the extreme similarity in temporal patterns across different segments, long-term time series $\mathbf{X}_{\text{his}} \in \mathbb{R}^{N \times P}$ often exhibits approximate low-rank characteristics:

$$\text{Rank}(\mathbf{X}_{\text{his}}) \ll \min(N, P) \tag{11}$$

**Generalization of TimeBase**  Specifically, in the context of TimeBase, the right singular vectors in $V^T$ of Eq(9) can be interpreted as candidate basis components $\mathbf{E} \in \mathbb{R}^{R \times P}$, where $R \ll \min(N, P)$ specifies the number of significant

components to be retained. Next, to reconstruct the time series data $\mathbf{X}_{\mathrm{his}}$ using these basis vectors, it can be achieved by a linear layer in a deep learning framework, which applies the following transformation: $\mathbf{X}_{\mathrm{his}}^T \approx \mathbf{E}^T W_{\mathrm{E}} + B$. Based on this, we can get $\mathbf{E}^T \approx (\mathbf{X}_{\mathrm{his}}^T - B) \times W_E^{\dagger}$, and the prediction is $\mathbf{X}_{\mathrm{pred}}^T = \mathbf{E}^T W_{\mathrm{pred}} + B_{\mathrm{pred}}$. The prediction error is defined as:

$$\mathbf{r} = \mathbf{X}_{\mathrm{pred}}^T - \mathbf{E}^T W_{\mathrm{pred}} - B_{\mathrm{pred}} \tag{12}$$

By ignoring the bias term $B_{\mathrm{pred}}$, the norm of the error is:

$$\|\mathbf{r}\|_2 = \|\mathbf{X}_{\mathrm{pred}} - W_{\mathrm{pred}}\mathbf{E}\|_2 \tag{13}$$

To derive the optimal coefficient matrix $W_{\mathrm{pred}}$, we need to solve the following optimization problem:

$$\begin{aligned}
&\min_{W_{\mathrm{pred}}} \|\mathbf{X}_{\mathrm{pred}} - W_{\mathrm{pred}}\mathbf{E}\|_2^2 \\
&= \min_{W_{\mathrm{pred}}} \mathrm{Tr}\left((\mathbf{X}_{\mathrm{pred}} - W_{\mathrm{pred}}\mathbf{E})(\mathbf{X}_{\mathrm{pred}} - W_{\mathrm{pred}}\mathbf{E})^T\right) \\
&= \min_{W_{\mathrm{pred}}} \mathrm{Tr}(\mathbf{X}_{\mathrm{pred}}\mathbf{X}_{\mathrm{pred}}^T) - 2\,\mathrm{Tr}(W_{\mathrm{pred}}\mathbf{E}\mathbf{X}_{\mathrm{pred}}^T) + \mathrm{Tr}(W_{\mathrm{pred}}\mathbf{E}\mathbf{E}^T W_{\mathrm{pred}}^T)
\end{aligned} \tag{14}$$

Next, we take the derivative:

$$\begin{aligned}
&\nabla_{W_{\mathrm{pred}}} \|\mathbf{X}_{\mathrm{pred}} - W_{\mathrm{pred}}\mathbf{E}\|_2^2 \\
&= \nabla_{W_{\mathrm{pred}}} \mathrm{Tr}(\mathbf{X}_{\mathrm{pred}}\mathbf{X}_{\mathrm{pred}}^T) - 2\,\mathrm{Tr}(W_{\mathrm{pred}}\mathbf{E}\mathbf{X}_{\mathrm{pred}}^T) + \mathrm{Tr}(W_{\mathrm{pred}}\mathbf{E}\mathbf{E}^T W_{\mathrm{pred}}^T) \\
&= -2\mathbf{X}_{\mathrm{pred}}\mathbf{E}^T + 2W_{\mathrm{pred}}\mathbf{E}\mathbf{E}^T
\end{aligned} \tag{15}$$

Setting the derivative equal to zero, we get:

$$W_{\mathrm{pred}} = \mathbf{X}_{\mathrm{pred}}\mathbf{E}^T(\mathbf{E}\mathbf{E}^T)^{-1} \tag{16}$$

Substituting the optimal coefficient $W_{\mathrm{pred}}$ into the error expression, we obtain the error:

$$\begin{aligned}
\mathbf{r} &= \mathbf{X}_{\mathrm{pred}} - \mathbf{X}_{\mathrm{pred}}\mathbf{E}^T(\mathbf{E}\mathbf{E}^T)^{-1}\mathbf{E} \\
&= \mathbf{X}_{\mathrm{pred}}(\mathbf{I} - \mathbf{E}^T(\mathbf{E}\mathbf{E}^T)^{-1}\mathbf{E})
\end{aligned} \tag{17}$$

Here, $\mathbf{P} = \mathbf{I} - \mathbf{E}^T(\mathbf{E}\mathbf{E}^T)^{-1}\mathbf{E}$ is a projection matrix. The norm property of the projection matrix can be bounded:

$$\mathbf{P} = \|\mathbf{I} - \mathbf{E}^T(\mathbf{E}\mathbf{E}^T)^{-1}\mathbf{E}\|_2 = 1 - \sigma_{\min}, \tag{18}$$

where $\sigma_{\min}$ represents the smallest singular value of $\mathbf{E}\mathbf{E}^T$. Since singular values are equivalent to eigenvalues in this context:

$$\sigma_{\min} = \lambda_{\min}(\mathbf{E}\mathbf{E}^T) \tag{19}$$

Next, based on the fact that the spectral norm of the projection matrix is bounded above by $\frac{1}{\lambda_{\min}(\mathbf{E}\mathbf{E}^T)}$ (Golub and Van Loan, 2013), and utilizing the inequality property of matrix norms, $\|\mathbf{A}\mathbf{B}\|_2 \leq \|\mathbf{A}\|_2\|\mathbf{B}\|_2$ (Horn and Johnson, 2012), we obtain:

$$\|\mathbf{r}\|_2 \leq \frac{1}{\lambda_{\min}(\mathbf{E}\mathbf{E}^T)}\|\mathbf{X}_{\mathrm{pred}}\|_2 \tag{20}$$

where $\lambda_{\min}(\mathbf{E}\mathbf{E}^T)$ denotes the smallest eigenvalue of the Gram matrix $\mathbf{E}\mathbf{E}^T$. This result highlights that the generalization capability of TimeBase to arbitrary time series relies on learning a well-represented basis matrix $\mathbf{E}$. If $\mathbf{E}$ exhibits a favorable eigenvalue distribution (i.e., $\lambda_{\min}(\mathbf{E}\mathbf{E}^T)$ is large), the upper bound on prediction error is lower, highlighting the importance of a learned high-quality basis components.

## C. Experiment Details

### C.1. Experiment Setups

We briefly describe the selected 10 state-of-the-art baselines as follows: **1) iTransformer** (Liu et al., 2024a) simply inverts the duties of the attention mechanism and the feed-forward network to encode each individual series into *variate tokens* and

for forecasting. **2) PatchTST** (Nie et al., 2023) is a strong versatile transformer baseline using channel-independence and patching. **3) TimesNet** (Wu et al., 2023) is a task-general foundational model for time series, reshaping 1-dim temporal data to 2-dim space and using 2-dim backbone to deal with the data. **4) FEDformer** (Zhou et al., 2022) introduces a frequency-enhanced decomposer to model seasonal-trend time series in an efficient manner. **5) Autoformer** (Wu et al., 2021) employs an auto-correlation mechanism and series decomposition block to improve long-sequence forecasting. **6) Informer** (Zhou et al., 2021) utilizes a sparse self-attention mechanism and a distilling operation to handle long time series more efficiently. **7) DLinear** (Zeng et al., 2023) is a simple linear-based model combined with a decomposition module. **8) FITS** (Xu et al., 2024) introduces an innovative method for time series forecasting using a complex-valued neural network, effectively capturing both the magnitude and phase of the data. This dual representation enables a more thorough and efficient analysis of time series signals. **9) SparseTSF** (Lin et al., 2024b) simplifies the time series forecasting process by down sampling the original sequences at fixed intervals. **10) TimeMixer** (Wang et al., 2024d) tackles the issue of different granularity levels in micro and macro series by proposing mixing blocks.

**Implementation Details**   We build TimeBase using PyTorch 1.13.0 (Paszke et al., 2019). The model is trained with the Adam optimizer (Kingma, 2014) with L2 loss over 30 epochs. After the first three epochs, a learning rate decay of 0.8 is applied, and early stopping is employed with a patience threshold of five epochs. In TimeBase, Channel Independence (CI) (Nie et al., 2023) is involved to simplify the multivariate forecasting process to univariate time series forecasting. Due to its highly simplistic design, TimeBase requires minimal hyperparameter tuning. The segment length $P$ is set to the natural period of the dataset (e.g., $P = 24$ for ETTh1), or respectively shorter when dealing with datasets that exhibit extremely long periods (e.g., $P = 4$ for Weather). We perform a grid search for TimeBase to find the optimal hyperparameters, specifically for the regularization parameter $\lambda_{orth} = [0.00, 0.04, 0.08, 0.12, 0.16, 0.20]$ to accommodate variances between datasets, as well as the learning rate between 0.01 and 0.5. The loss function is MSE. To enhance the reliability of our results, we re-run baselines in an uniform and fair setting. For SparseTSF, PatchTST, DLinear, FITS, TimeMixer, Fedformer, and TimesNet, we utilized their official code repositories. For Autoformer and Informer, we leverage the code provided in the official DLinear repository to run these models. To ensure a fair comparison with standard LTSF baselines such as  (Xu et al., 2024; Lin et al., 2024b), which utilize a uniform input length of 720, we also adopt an input length of 720 for all models. It is important to note that we have corrected the bug involving `test_loader` where `drop_last=True` during testing on the test set, ensuring that `drop_last=False` is used instead, which ensures a fair comparison in LTSF(Qiu et al., 2024).

### C.2. Additional Results

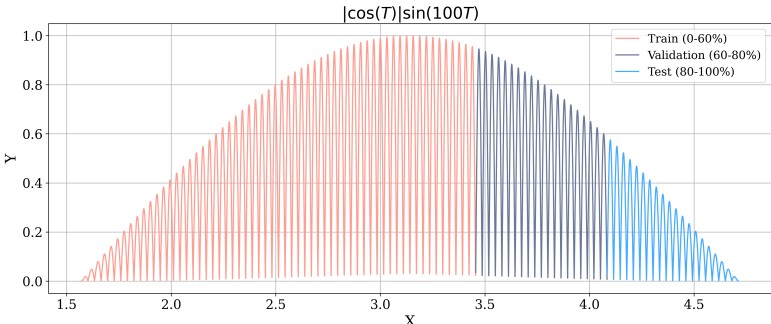

*Figure 7.* Visualization of the synthetic data generated using the equation $Y = |\cos(X)|\sin(100X)$. The dataset comprises 5000 samples, which are split into training (60%), validation (20%), and testing (20%) sets.

**Effectiveness on Synthetic Data**   The synthetic dataset derived from the function $Y = |\cos(X)|\sin(100X)$ is shown in Figure  7, and it provides a controlled and challenging benchmark to evaluate the forecasting performance of TimeBase. The data was sampled with high frequency to capture intricate oscillations, divided into training, validation, and testing sets in a 6:2:2 ratio. The basis number is set as 6 for TimeBase to achieve efficient basis extraction. TimeBase consistently demonstrated superior accuracy and efficiency across varying prediction lengths. For a forecasting length of 100, it achieved an MSE of 0.007 and an MAE of 0.070, utilizing only 0.19K parameters and 0.077M MACs, substantially less than its competitors. At a forecasting length of 200, TimeBase maintained robust accuracy (MSE = 0.010, MAE = 0.082) while still requiring minimal resources (0.23K parameters, 0.093M MACs). For a length of 300, it continued to excel in efficiency and delivered competitive accuracy (MSE = 0.013, MAE = 0.094), outperforming DLinear and approaching PatchTST's

*Table 7.* Performance of TimeBase on synthetic data for a forecasting length of 100, 200, 300.

| $L$ | Model | MSE | MAE | MACs | Params |
|---|---|---|---|---|---|
| 100 | TimeBase | 0.007 | 0.070 | 0.077M | 0.19K |
|  | DLinear | 0.015 | 0.081 | 0.1M | 100K |
|  | PatchTST | 0.011 | 0.085 | 1.28M | 219K |
| 200 | TimeBase | 0.010 | 0.082 | 0.093M | 0.23K |
|  | DLinear | 0.021 | 0.124 | 0.2M | 200K |
|  | PatchTST | 0.017 | 0.083 | 1.48M | 420K |
| 300 | TimeBase | 0.013 | 0.094 | 0.099M | 0.26K |
|  | DLinear | 0.039 | 0.180 | 0.3M | 300K |
|  | PatchTST | 0.023 | 0.136 | 1.69M | 622K |

accuracy but at a fraction of the computational cost.

## C.3. Ablation Study

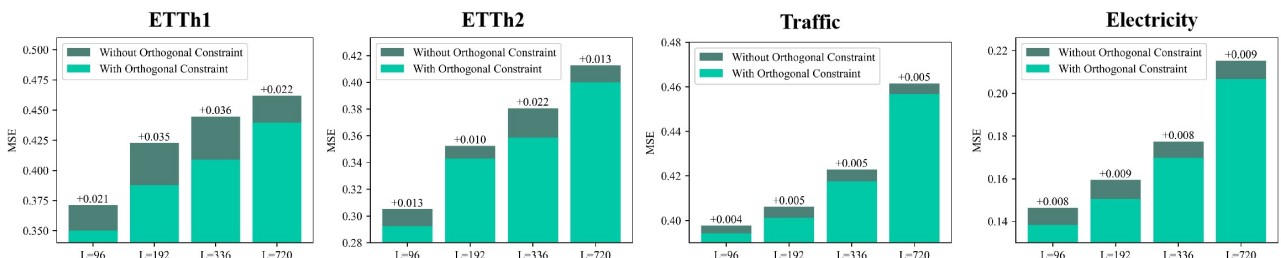

*Figure 8.* MSE comparison with and without orthogonal constraint across different prediction lengths for Traffic, Electricity, ETTh1, and ETTh2 datasets.

Figure 8 illustrates the MSE results for various prediction lengths, both with and without the orthogonal constraint. The inclusion of the orthogonal constraint consistently leads to improvements across all datasets, with gains up to 0.036 in MSE. This indicates that the orthogonal constraint helps the model learn more representative basis components, enhancing both the model's representational capacity and predictive performance. The positive impact across multiple datasets and forecasting horizons demonstrates the value of incorporating the orthogonal constraint into the training process.

## C.4. Segment Length Analysis

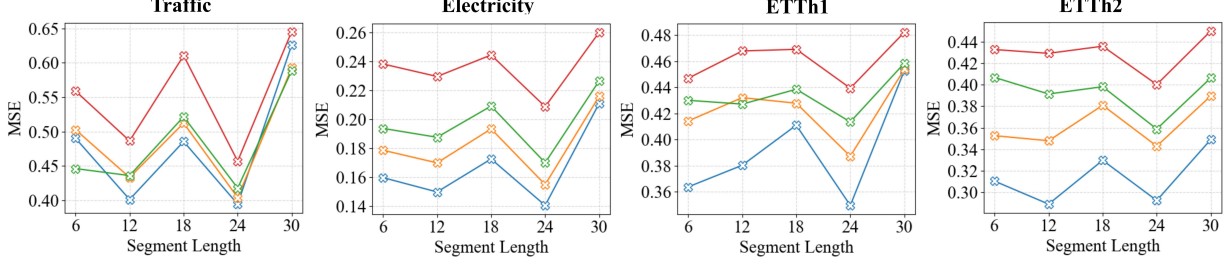

*Figure 9.* MSE results for different segment lengths ($P = [6, 12, 18, 24, 30]$) across various prediction lengths on Traffic, Electricity, ETTh1, and ETTh2 datasets.

This section explores the impact of segment length on the forecasting performance of TimeBase across the Traffic, Electricity, ETTh1, and ETTh2 datasets. The analysis evaluates how varying the length of sub-sequence segments, denoted as $P$, affects prediction accuracy for different forecasting horizons. Figure 9 shows that across all datasets and forecasting horizons, the best performance is consistently achieved when the segment length is set to $P = 24$. In contrast, shorter or longer segment lengths ($P = 6, 18, 30$) result in noticeably higher MSE values, indicating suboptimal performance. This suggests that

the choice of segment length significantly affects the model's ability to capture temporal patterns effectively. The superior performance at $P = 24$ highlights the importance of aligning the segment length with the inherent periodicity of the data. Deviation from this optimal segment length reduces the model's capacity to accurately represent the underlying time series dynamics, thus leading to a degradation in forecasting accuracy. This analysis underscores the necessity of selecting an appropriate segment length that corresponds to the temporal nature of the data. The findings suggest that segmenting the time series into periods of $P = 24$ yields the most representative and predictive sub-sequences, enhancing overall forecasting performance.

## C.5. Performance on Further Prediction

To better illustrate its strengths in long-term time series forecasting, we extended the maximum prediction horizon beyond 720 to include 1080, 1440, and 1800 time steps. We compared its performance against well-established LTSF models, such as iTransformer and DLinear, across multiple datasets (ETTh1, ETTh2, and Electricity). As shown in Table 8, the results underscore that TimeBase consistently outperforms these models in ultra-long-term forecasting tasks. It achieves this while maintaining linear growth in model complexity (measured by parameters, MACs), demonstrating scalability as the prediction length increases. Specifically, TimeBase not only yields lower Mean Squared Error (MSE) values but also achieves these results with significantly fewer parameters and MACs compared to its counterparts. For example, in the ETTh1 dataset at a prediction length of 1800, TimeBase achieves an MSE of 0.714 with only 0.7K parameters and 0.1M MACs. In contrast, iTransformer and DLinear exhibit higher MSEs of 0.812 and 0.796, respectively, while using 523.9K and 2.59M parameters, and 6.78M and 18.15M MACs. Similar trends are observed across the ETTh2 and Electricity datasets, where TimeBase demonstrates robust accuracy and efficiency advantages. These findings validate TimeBase's effectiveness in ultra-long-term forecasting tasks, particularly when resource efficiency is critical. Moreover, the linear growth in computational cost ensures its feasibility for deployment on edge devices.

*Table 8.* Further prediction length on Electricity, ETTh2 and ETTh1. The input length is set as 720 for all models.

| | L | | 1080 | | | 1440 | | | 1800 | |
|---|---|---|---|---|---|---|---|---|---|---|
| | Metric | MSE | Param | MAC | MSE | Param | MAC | MSE | Param | MAC |
| ELC. | **TimeBase** | **0.234** | **0.5 K** | **3.47 M** | **0.264** | **0.6 K** | **4.16 M** | **0.295** | **0.7 K** | **4.85 M** |
| | iTrans. | 0.253 | 5.65 M | 1.85 G | 0.272 | 5.84 M | 1.91 G | 0.325 | 6.03 M | 1.97 G |
| | DLinear | 0.255 | 1.6 M | 499.45 M | 0.290 | 2.1 M | 665.86 M | 0.321 | 2.59 M | 832.26 M |
| ETTh2 | **TimeBase** | **0.478** | **0.5 K** | **0.07 M** | **0.543** | **0.6 K** | **0.09 M** | **0.552** | **0.7 K** | **0.1 M** |
| | iTrans. | 0.501 | 431.1 K | 5.58 M | 0.575 | 477.4 K | 6.18 M | 0.597 | 523.9 K | 6.78 M |
| | DLinear | 0.583 | 1.6 M | 10.89 M | 0.672 | 2.1 M | 14.52 M | 0.652 | 2.59 M | 18.15 M |
| ETTh1 | **TimeBase** | **0.551** | **0.5 K** | **0.07 M** | **0.636** | **0.6 K** | **0.09 M** | **0.714** | **0.7 K** | **0.1 M** |
| | iTrans. | 0.602 | 431.1 K | 5.58 M | 0.708 | 477.4 K | 6.18 M | 0.812 | 523.9 K | 6.78 M |
| | DLinear | 0.582 | 160 M | 10.89 M | 0.693 | 2.1 M | 14.52 M | 0.796 | 2.59 M | 18.15 M |

## C.6. Extension to Multi-Seasonality

In time series forecasting, many real-world datasets exhibit multiple seasonalities, which are crucial for accurate modeling and prediction. For example, traffic data often contains both daily and weekly seasonal patterns, which can significantly influence forecasting accuracy. TimeBase, initially designed for univariate time series forecasting, can be extended to handle such multi-seasonal data by learning distinct period bases for each individual seasonality and combining their outputs to generate more accurate predictions. This extension allows TimeBase to model complex periodic patterns while retaining its minimalistic architecture. Mathematically, the multi-seasonal extension of TimeBase can be formulated as follows:

$$\text{MSTimeBase} = \sum_i \text{TimeBase}(\mathbf{X}; \mathbf{P} = p_i) \tag{21}$$

where $\mathbf{X}$ represents the input data and $\mathbf{P} = p_i$ denotes the different period bases corresponding to each seasonality. By learning multiple period bases ($p \in [24, 168]$ hours, for example), the model can capture both short-term and long-term seasonal patterns and combine them to enhance the accuracy of the forecast. To evaluate this extension, we applied the multi-seasonality approach to the Traffic dataset, considering both daily and weekly seasonalities. The results, summarized in Table 9, demonstrate that incorporating multiple seasonalities into TimeBase improves the prediction performance with only a modest increase in computational cost and model complexity. Specifically, the model's Mean Squared Error (MSE)

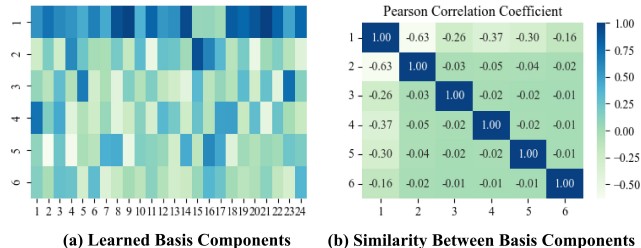

(a) Learned Basis Components    (b) Similarity Between Basis Components

*Figure 10.* Visualization of the learned basis components on the Electricity dataset and the corresponding Pearson correlation coefficients.

and Mean Absolute Error (MAE) improve when compared to the original TimeBase model, despite only a slight increase in the number of parameters and computational cost. This extension underscores the versatility and power of TimeBase in dealing with complex, multi-seasonal patterns, making it suitable for a wide range of LTSF tasks. The ability to extend TimeBase while maintaining its lightweight nature reflects its potential for scalable deployment in real-world applications, where seasonalities often vary and need to be captured for accurate forecasting.

*Table 9.* Performance of TimeBase extended to multi-seasonality. The prediction length is 720 for Traffic dataset.

| Model | MSE | MAE | MACs | Params | $R$ |
|---|---|---|---|---|---|
| iTransformer | **0.450** | 0.313 | 1.01 G | 11.61 M | / |
| TimeBase | 0.456 | 0.301 | **9.93 M** | 0.51 K | 8 |
| MSTimeBase | **0.450** | **0.295** | 16.76 M | **0.49 K** | 6 |

## C.7. Basis Visualization

Figure 10 displays the basis components extracted by TimeBase from the Electricity dataset, along with the Pearson correlation matrix between them. The results show that the Pearson correlation coefficients between most basis components are close to zero, indicating low correlation among them. This suggests that TimeBase is capable of extracting distinct and representative basis components from the approximate low-rank structure of long-term time series data. By identifying these representative basis components, TimeBase can perform segment-level forecasting using a compact set of basis components. This approach significantly reduces the number of parameters required by the model while maintaining competitive forecasting performance. The ability to leverage such compact representations is key to TimeBase's parameter efficiency and contributes to its strong performance on long-term time series forecasting tasks.

## C.8. Full Results

*Table 10.* Full Long-term time series forecasting results in normal scale datasets, comparing TimeBase with other baselines. Input lengths of all models are set to 720. Top 2 results are highlighted in bold.

| Methods | | TimeBase (ours) MSE | MAE | SparseTSF (2024b) MSE | MAE | TimeMixer (2024d) MSE | MAE | FITS (2024) MSE | MAE | iTransformer (2024a) MSE | MAE | DLinear (2023) MSE | MAE | PatchTST (2023) MSE | MAE | TimesNet (2023) MSE | MAE | FEDformer (2022) MSE | MAE | Autoformer (2021) MSE | MAE | Informer (2021) MSE | MAE |
|---|---|---|---|---|---|---|---|---|---|---|---|---|---|---|---|---|---|---|---|---|---|---|---|
| ETTh1 | 96 | 0.349 | 0.384 | 0.362 | 0.389 | 0.410 | 0.441 | 0.380 | 0.402 | 0.389 | 0.421 | 0.378 | 0.402 | 0.377 | 0.408 | 0.437 | 0.454 | 0.485 | 0.500 | 0.555 | 0.558 | 1.269 | 0.855 |
| | 192 | 0.387 | 0.410 | 0.404 | 0.412 | 0.448 | 0.465 | 0.415 | 0.424 | 0.424 | 0.446 | 0.415 | 0.425 | 0.413 | 0.431 | 0.456 | 0.469 | 0.481 | 0.498 | 0.599 | 0.575 | 1.487 | 0.943 |
| | 336 | 0.408 | 0.418 | 0.435 | 0.428 | 0.482 | 0.490 | 0.439 | 0.439 | 0.456 | 0.469 | 0.449 | 0.449 | 0.436 | 0.446 | 0.494 | 0.494 | 0.522 | 0.521 | 0.853 | 0.702 | 1.544 | 0.945 |
| | 720 | 0.439 | 0.446 | 0.426 | 0.448 | 0.475 | 0.500 | 0.433 | 0.457 | 0.545 | 0.532 | 0.507 | 0.517 | 0.455 | 0.475 | 0.632 | 0.578 | 0.604 | 0.575 | 0.899 | 0.730 | 1.481 | 0.975 |
| ETTh2 | 96 | 0.292 | 0.345 | 0.294 | 0.346 | 0.315 | 0.380 | 0.271 | 0.336 | 0.305 | 0.361 | 0.294 | 0.360 | 0.276 | 0.339 | 0.349 | 0.403 | 0.401 | 0.451 | 0.541 | 0.559 | 5.189 | 1.812 |
| | 192 | 0.339 | 0.387 | 0.340 | 0.377 | 0.383 | 0.415 | 0.332 | 0.374 | 0.405 | 0.421 | 0.412 | 0.437 | 0.342 | 0.385 | 0.500 | 0.488 | 0.425 | 0.464 | 1.207 | 0.866 | 6.514 | 2.011 |
| | 336 | 0.358 | 0.410 | 0.360 | 0.398 | 0.385 | 0.438 | 0.355 | 0.396 | 0.411 | 0.436 | 0.471 | 0.478 | 0.364 | 0.405 | 0.445 | 0.465 | 0.427 | 0.471 | 0.825 | 0.719 | 5.284 | 1.859 |
| | 720 | 0.400 | 0.448 | 0.383 | 0.425 | 0.432 | 0.471 | 0.378 | 0.423 | 0.448 | 0.470 | 0.740 | 0.609 | 0.395 | 0.434 | 0.438 | 0.465 | 0.462 | 0.493 | 1.772 | 1.062 | 4.955 | 1.884 |
| ETTm1 | 96 | 0.311 | 0.351 | 0.314 | 0.359 | 0.332 | 0.384 | 0.313 | 0.357 | 0.315 | 0.369 | 0.314 | 0.350 | 0.298 | 0.352 | 0.359 | 0.391 | 0.406 | 0.441 | 0.455 | 0.464 | 0.632 | 0.574 |
| | 192 | 0.338 | 0.371 | 0.348 | 0.376 | 0.355 | 0.398 | 0.339 | 0.369 | 0.349 | 0.388 | 0.347 | 0.381 | 0.335 | 0.373 | 0.368 | 0.398 | 0.450 | 0.477 | 0.562 | 0.514 | 1.131 | 0.802 |
| | 336 | 0.364 | 0.386 | 0.368 | 0.386 | 0.386 | 0.416 | 0.367 | 0.385 | 0.381 | 0.409 | 0.367 | 0.387 | 0.366 | 0.394 | 0.429 | 0.438 | 0.436 | 0.466 | 0.737 | 0.608 | 1.391 | 0.923 |
| | 720 | 0.413 | 0.414 | 0.419 | 0.413 | 0.452 | 0.457 | 0.417 | 0.417 | 0.437 | 0.439 | 0.415 | 0.415 | 0.420 | 0.421 | 0.477 | 0.474 | 0.462 | 0.479 | 0.503 | 0.502 | 1.397 | 0.973 |
| ETTm2 | 96 | 0.162 | 0.256 | 0.167 | 0.259 | 0.192 | 0.285 | 0.166 | 0.256 | 0.179 | 0.274 | 0.163 | 0.257 | 0.165 | 0.260 | 0.200 | 0.288 | 0.339 | 0.406 | 0.325 | 0.391 | 1.870 | 1.002 |
| | 192 | 0.218 | 0.293 | 0.219 | 0.297 | 0.253 | 0.329 | 0.218 | 0.293 | 0.239 | 0.314 | 0.223 | 0.304 | 0.219 | 0.298 | 0.274 | 0.337 | 0.397 | 0.452 | 0.369 | 0.414 | 2.807 | 1.314 |
| | 336 | 0.270 | 0.328 | 0.271 | 0.330 | 0.307 | 0.362 | 0.271 | 0.328 | 0.309 | 0.356 | 0.291 | 0.355 | 0.268 | 0.333 | 0.340 | 0.362 | 0.449 | 0.491 | 0.418 | 0.452 | 4.442 | 1.661 |
| | 720 | 0.352 | 0.380 | 0.353 | 0.380 | 0.380 | 0.412 | 0.352 | 0.380 | 0.387 | 0.407 | 0.407 | 0.433 | 0.352 | 0.386 | 0.384 | 0.407 | 0.451 | 0.499 | 0.612 | 0.594 | 5.258 | 1.914 |
| Weather | 96 | 0.146 | 0.198 | 0.174 | 0.231 | 0.163 | 0.223 | 0.176 | 0.232 | 0.159 | 0.212 | 0.174 | 0.242 | 0.149 | 0.199 | 0.176 | 0.234 | 0.289 | 0.342 | 0.323 | 0.389 | 0.283 | 0.361 |
| | 192 | 0.185 | 0.241 | 0.216 | 0.267 | 0.201 | 0.254 | 0.216 | 0.268 | 0.203 | 0.252 | 0.215 | 0.277 | 0.193 | 0.243 | 0.219 | 0.270 | 0.340 | 0.394 | 0.389 | 0.423 | 0.445 | 0.461 |
| | 336 | 0.236 | 0.281 | 0.260 | 0.299 | 0.258 | 0.300 | 0.261 | 0.299 | 0.253 | 0.291 | 0.262 | 0.319 | 0.240 | 0.281 | 0.277 | 0.311 | 0.370 | 0.408 | 0.497 | 0.495 | 0.587 | 0.526 |
| | 720 | 0.309 | 0.331 | 0.325 | 0.345 | 0.329 | 0.348 | 0.325 | 0.346 | 0.317 | 0.337 | 0.319 | 0.359 | 0.312 | 0.334 | 0.344 | 0.356 | 0.420 | 0.421 | 0.573 | 0.520 | 0.953 | 0.703 |
| Electricity | 96 | 0.139 | 0.231 | 0.139 | 0.239 | 0.142 | 0.247 | 0.147 | 0.235 | 0.135 | 0.233 | 0.141 | 0.244 | 0.141 | 0.240 | 0.202 | 0.308 | 0.226 | 0.341 | 0.225 | 0.334 | 0.395 | 0.460 |
| | 192 | 0.153 | 0.245 | 0.155 | 0.250 | 0.164 | 0.275 | 0.159 | 0.256 | 0.155 | 0.253 | 0.155 | 0.258 | 0.156 | 0.256 | 0.218 | 0.322 | 0.220 | 0.336 | 0.223 | 0.332 | 0.405 | 0.460 |
| | 336 | 0.169 | 0.262 | 0.171 | 0.265 | 0.171 | 0.260 | 0.169 | 0.260 | 0.169 | 0.267 | 0.170 | 0.275 | 0.172 | 0.267 | 0.232 | 0.332 | 0.224 | 0.337 | 0.233 | 0.341 | 0.404 | 0.460 |
| | 720 | 0.207 | 0.294 | 0.208 | 0.300 | 0.209 | 0.313 | 0.214 | 0.302 | 0.204 | 0.301 | 0.209 | 0.309 | 0.208 | 0.299 | 0.299 | 0.375 | 0.271 | 0.378 | 0.261 | 0.364 | 0.429 | 0.477 |
| Traffic | 96 | 0.394 | 0.267 | 0.389 | 0.268 | 0.396 | 0.294 | 0.402 | 0.275 | 0.374 | 0.273 | 0.396 | 0.272 | 0.363 | 0.250 | 0.605 | 0.325 | 0.664 | 0.431 | 0.668 | 0.401 | 0.829 | 0.487 |
| | 192 | 0.403 | 0.271 | 0.399 | 0.272 | 0.404 | 0.295 | 0.419 | 0.286 | 0.393 | 0.283 | 0.404 | 0.275 | 0.382 | 0.258 | 0.627 | 0.340 | 0.613 | 0.382 | 0.703 | 0.439 | 0.902 | 0.525 |
| | 336 | 0.417 | 0.278 | 0.417 | 0.279 | 0.419 | 0.302 | 0.423 | 0.292 | 0.409 | 0.292 | 0.417 | 0.283 | 0.399 | 0.268 | 0.631 | 0.349 | 0.612 | 0.379 | 0.666 | 0.421 | 0.949 | 0.545 |
| | 720 | 0.456 | 0.298 | 0.449 | 0.299 | 0.458 | 0.309 | 0.459 | 0.311 | 0.450 | 0.314 | 0.457 | 0.310 | 0.432 | 0.289 | 0.700 | 0.371 | 0.664 | 0.410 | 0.697 | 0.424 | 1.430 | 0.793 |
| AQShunyi | 96 | 0.629 | 0.489 | 0.751 | 0.553 | 0.703 | 0.534 | 0.759 | 0.555 | 0.686 | 0.508 | 0.751 | 0.579 | 0.643 | 0.476 | 0.759 | 0.560 | 1.247 | 0.819 | 1.394 | 0.931 | 1.221 | 0.864 |
| | 192 | 0.661 | 0.500 | 0.774 | 0.551 | 0.720 | 0.524 | 0.774 | 0.553 | 0.728 | 0.520 | 0.771 | 0.571 | 0.692 | 0.501 | 0.785 | 0.557 | 1.218 | 0.813 | 1.394 | 0.872 | 1.595 | 0.951 |
| | 336 | 0.683 | 0.510 | 0.752 | 0.544 | 0.747 | 0.546 | 0.755 | 0.544 | 0.732 | 0.529 | 0.758 | 0.580 | 0.695 | 0.514 | 0.802 | 0.566 | 1.071 | 0.742 | 1.438 | 0.900 | 1.699 | 0.957 |
| | 720 | 0.725 | 0.523 | 0.763 | 0.537 | 0.772 | 0.541 | 0.763 | 0.538 | 0.744 | 0.524 | 0.748 | 0.559 | 0.732 | 0.520 | 0.807 | 0.554 | 0.985 | 0.655 | 1.344 | 0.809 | 2.236 | 1.094 |
| AQWan | 96 | 0.726 | 0.477 | 0.758 | 0.486 | 0.762 | 0.484 | 0.795 | 0.510 | 0.798 | 0.511 | 0.835 | 0.531 | 0.735 | 0.467 | 0.828 | 0.527 | 0.865 | 0.523 | 0.895 | 0.592 | 1.027 | 0.699 |
| | 192 | 0.761 | 0.492 | 0.851 | 0.542 | 0.779 | 0.492 | 0.765 | 0.485 | 0.785 | 0.501 | 0.884 | 0.558 | 0.792 | 0.500 | 0.807 | 0.510 | 1.001 | 0.603 | 0.935 | 0.600 | 1.293 | 0.834 |
| | 336 | 0.783 | 0.502 | 0.810 | 0.531 | 0.857 | 0.559 | 0.861 | 0.564 | 0.818 | 0.537 | 0.842 | 0.550 | 0.804 | 0.525 | 0.810 | 0.529 | 0.893 | 0.569 | 0.990 | 0.666 | 1.017 | 0.711 |
| | 720 | 0.847 | 0.524 | 0.883 | 0.546 | 0.891 | 0.549 | 0.831 | 0.517 | 0.971 | 0.604 | 0.972 | 0.600 | 0.909 | 0.561 | 0.982 | 0.606 | 0.967 | 0.566 | 1.185 | 0.759 | 1.156 | 0.731 |
| CzeLan | 96 | 0.177 | 0.224 | 0.204 | 0.263 | 0.182 | 0.233 | 0.208 | 0.268 | 0.202 | 0.259 | 0.187 | 0.239 | 0.184 | 0.236 | 0.204 | 0.262 | 0.208 | 0.260 | 0.251 | 0.334 | 0.293 | 0.384 |
| | 192 | 0.211 | 0.250 | 0.227 | 0.274 | 0.216 | 0.259 | 0.242 | 0.292 | 0.233 | 0.282 | 0.226 | 0.271 | 0.214 | 0.257 | 0.256 | 0.308 | 0.258 | 0.307 | 0.285 | 0.360 | 0.329 | 0.399 |
| | 336 | 0.243 | 0.276 | 0.239 | 0.305 | 0.246 | 0.312 | 0.247 | 0.316 | 0.239 | 0.305 | 0.238 | 0.301 | 0.229 | 0.290 | 0.274 | 0.347 | 0.299 | 0.365 | 0.263 | 0.354 | 0.366 | 0.476 |
| | 720 | 0.269 | 0.297 | 0.292 | 0.325 | 0.291 | 0.322 | 0.290 | 0.338 | 0.305 | 0.340 | 0.315 | 0.349 | 0.288 | 0.319 | 0.299 | 0.332 | 0.300 | 0.324 | 0.357 | 0.423 | 0.399 | 0.446 |
| ZafNoo | 96 | 0.435 | 0.407 | 0.435 | 0.439 | 0.410 | 0.411 | 0.458 | 0.461 | 0.442 | 0.447 | 0.423 | 0.423 | 0.412 | 0.413 | 0.443 | 0.444 | 0.484 | 0.473 | 0.548 | 0.566 | 0.651 | 0.672 |
| | 192 | 0.470 | 0.435 | 0.548 | 0.509 | 0.473 | 0.437 | 0.506 | 0.468 | 0.560 | 0.522 | 0.534 | 0.494 | 0.518 | 0.479 | 0.572 | 0.529 | 0.579 | 0.512 | 0.738 | 0.712 | 0.745 | 0.730 |
| | 336 | 0.511 | 0.453 | 0.525 | 0.474 | 0.576 | 0.520 | 0.528 | 0.477 | 0.536 | 0.486 | 0.596 | 0.537 | 0.528 | 0.477 | 0.579 | 0.523 | 0.566 | 0.499 | 0.624 | 0.564 | 0.662 | 0.601 |
| | 720 | 0.565 | 0.484 | 0.617 | 0.540 | 0.560 | 0.487 | 0.590 | 0.515 | 0.623 | 0.546 | 0.606 | 0.527 | 0.581 | 0.506 | 0.587 | 0.511 | 0.776 | 0.671 | 0.758 | 0.698 | 0.847 | 0.783 |
| Exchange | 96 | 0.087 | 0.207 | 0.083 | 0.211 | 0.090 | 0.229 | 0.094 | 0.241 | 0.096 | 0.244 | 0.096 | 0.243 | 0.083 | 0.211 | 0.088 | 0.224 | 0.092 | 0.224 | 0.119 | 0.316 | 0.106 | 0.276 |
| | 192 | 0.185 | 0.306 | 0.172 | 0.317 | 0.167 | 0.305 | 0.167 | 0.307 | 0.176 | 0.325 | 0.169 | 0.310 | 0.168 | 0.307 | 0.173 | 0.317 | 0.208 | 0.355 | 0.237 | 0.444 | 0.252 | 0.482 |
| | 336 | 0.355 | 0.436 | 0.308 | 0.395 | 0.342 | 0.437 | 0.319 | 0.409 | 0.312 | 0.399 | 0.344 | 0.438 | 0.324 | 0.413 | 0.362 | 0.462 | 0.405 | 0.512 | 0.437 | 0.561 | 0.536 | 0.700 |
| | 720 | 0.608 | 0.619 | 0.701 | 0.716 | 0.721 | 0.734 | 0.629 | 0.643 | 0.659 | 0.635 | 0.637 | 0.648 | 0.659 | 0.670 | 0.776 | 0.671 | 0.813 | 0.795 | 0.994 | 1.035 | 0.835 | 0.865 |
| METR-LA | 96 | 1.028 | 0.655 | 1.056 | 0.667 | 1.051 | 0.657 | 1.047 | 0.661 | 1.023 | 0.640 | 1.096 | 0.685 | 1.053 | 0.658 | 1.206 | 0.755 | 1.440 | 0.882 | 1.443 | 0.908 | 1.666 | 1.088 |
| | 192 | 1.112 | 0.711 | 1.232 | 0.730 | 1.163 | 0.689 | 1.295 | 0.772 | 1.196 | 0.713 | 1.330 | 0.788 | 1.134 | 0.672 | 1.359 | 0.806 | 1.478 | 0.839 | 1.366 | 0.830 | 1.452 | 0.904 |
| | 336 | 1.317 | 0.771 | 1.356 | 0.772 | 1.304 | 0.737 | 1.309 | 0.743 | 1.320 | 0.750 | 1.353 | 0.764 | 1.250 | 0.706 | 1.376 | 0.778 | 1.555 | 0.860 | 1.525 | 0.883 | 2.036 | 1.209 |
| | 720 | 1.558 | 0.877 | 1.562 | 0.885 | 1.564 | 0.879 | 1.534 | 0.865 | 1.474 | 0.834 | 1.423 | 0.800 | 1.404 | 0.789 | 1.619 | 0.910 | 1.725 | 0.921 | 2.221 | 1.277 | 2.420 | 1.384 |
| PM2.5 | 96 | 0.432 | 0.435 | 0.475 | 0.482 | 0.445 | 0.449 | 0.495 | 0.501 | 0.466 | 0.474 | 0.437 | 0.441 | 0.450 | 0.454 | 0.462 | 0.467 | 0.560 | 0.550 | 0.596 | 0.615 | 0.566 | 0.593 |
| | 192 | 0.429 | 0.432 | 0.458 | 0.465 | 0.479 | 0.483 | 0.455 | 0.460 | 0.505 | 0.511 | 0.454 | 0.458 | 0.461 | 0.465 | 0.452 | 0.456 | 0.480 | 0.474 | 0.567 | 0.614 | 0.670 | 0.678 |
| | 336 | 0.426 | 0.433 | 0.478 | 0.482 | 0.473 | 0.475 | 0.450 | 0.453 | 0.461 | 0.467 | 0.449 | 0.451 | 0.475 | 0.477 | 0.490 | 0.492 | 0.586 | 0.555 | 0.671 | 0.684 | 0.749 | 0.793 |
| | 720 | 0.425 | 0.448 | 0.456 | 0.479 | 0.464 | 0.487 | 0.440 | 0.461 | 0.489 | 0.514 | 0.457 | 0.479 | 0.448 | 0.470 | 0.461 | 0.484 | 0.612 | 0.609 | 0.652 | 0.686 | 0.561 | 0.589 |
| Solar | 96 | 0.192 | 0.239 | 0.205 | 0.241 | 0.232 | 0.271 | 0.218 | 0.257 | 0.217 | 0.255 | 0.200 | 0.234 | 0.205 | 0.239 | 0.243 | 0.284 | 0.231 | 0.254 | 0.279 | 0.339 | 0.308 | 0.377 |
| | 192 | 0.213 | 0.252 | 0.215 | 0.265 | 0.238 | 0.293 | 0.207 | 0.256 | 0.208 | 0.257 | 0.239 | 0.294 | 0.227 | 0.280 | 0.253 | 0.311 | 0.257 | 0.301 | 0.337 | 0.418 | 0.332 | 0.422 |
| | 336 | 0.222 | 0.261 | 0.213 | 0.276 | 0.234 | 0.301 | 0.229 | 0.295 | 0.238 | 0.309 | 0.231 | 0.298 | 0.225 | 0.290 | 0.222 | 0.287 | 0.260 | 0.319 | 0.263 | 0.352 | 0.314 | 0.421 |
| | 720 | 0.235 | 0.264 | 0.232 | 0.272 | 0.273 | 0.319 | 0.261 | 0.307 | 0.270 | 0.319 | 0.237 | 0.277 | 0.249 | 0.291 | 0.254 | 0.298 | 0.260 | 0.290 | 0.345 | 0.413 | 0.355 | 0.424 |
| Temp | 96 | 0.142 | 0.286 | 0.149 | 0.300 | 0.158 | 0.316 | 0.168 | 0.338 | 0.155 | 0.310 | 0.165 | 0.329 | 0.145 | 0.291 | 0.155 | 0.309 | 0.173 | 0.338 | 0.201 | 0.411 | 0.227 | 0.459 |
| | 192 | 0.155 | 0.305 | 0.168 | 0.323 | 0.163 | 0.312 | 0.181 | 0.351 | 0.168 | 0.324 | 0.174 | 0.335 | 0.166 | 0.319 | 0.187 | 0.358 | 0.179 | 0.326 | 0.246 | 0.491 | 0.281 | 0.564 |
| | 336 | 0.176 | 0.323 | 0.220 | 0.373 | 0.249 | 0.421 | 0.223 | 0.381 | 0.232 | 0.395 | 0.246 | 0.417 | 0.221 | 0.374 | 0.236 | 0.400 | 0.250 | 0.412 | 0.312 | 0.554 | 0.349 | 0.606 |
| | 720 | 0.274 | 0.437 | 0.642 | 0.654 | 0.637 | 0.643 | 0.689 | 0.699 | 0.652 | 0.663 | 0.584 | 0.590 | 0.632 | 0.639 | 0.636 | 0.642 | 0.769 | 0.747 | 0.859 | 0.893 | 1.026 | 1.101 |
| Wind | 96 | 0.785 | 0.626 | 0.796 | 0.589 | 0.908 | 0.669 | 0.787 | 0.581 | 0.824 | 0.609 | 0.824 | 0.607 | 0.802 | 0.591 | 0.892 | 0.657 | 1.008 | 0.718 | 1.110 | 0.847 | 1.132 | 0.836 |
| | 192 | 0.902 | 0.690 | 0.949 | 0.698 | 1.040 | 0.764 | 0.910 | 0.674 | 1.054 | 0.779 | 0.961 | 0.707 | 0.944 | 0.694 | 1.008 | 0.741 | 1.256 | 0.876 | 1.218 | 0.954 | 1.540 | 1.188 |
| | 336 | 0.997 | 0.737 | 1.182 | 0.851 | 1.039 | 0.747 | 1.100 | 0.795 | 1.025 | 0.742 | 1.022 | 0.734 | 1.043 | 0.750 | 1.185 | 0.853 | 1.375 | 0.923 | 1.209 | 0.935 | 1.399 | 1.015 |
| | 720 | 1.075 | 0.782 | 1.127 | 0.808 | 1.189 | 0.851 | 1.295 | 0.932 | 1.196 | 0.863 | 1.266 | 0.907 | 1.084 | 0.776 | 1.320 | 0.946 | 1.281 | 0.846 | 1.410 | 1.053 | 1.453 | 1.094 |

