# OpenReview forum: "TimeBase: The Power of Minimalism in Efficient Long-term Time Series Forecasting"
_ICML.cc/2025/Conference — ICML 2025 spotlightposter_

### Official Review · Reviewer_Pc2P · 2025-03-03

**Overall Recommendation:** 4

**Summary:**

This article presents a lightweight model requiring only 0.39k parameters, providing an extremely effective method for time-series forecasting. Extensive experiments have been conducted to verify the effectiveness of such a proposal.

**Claims And Evidence:**

The claims made in the submission are supported by sufficient experiments.

**Essential References Not Discussed:**

No

**Ethical Review Concerns:**

No ethical concerns.

**Experimental Designs Or Analyses:**

I focused on the section discussing the model's efficiency. In Section 4.3, the author provides detailed efficiency metrics, showing that both runtime and actual computational load are significantly lower than existing lightweight models, demonstrating that TimeBase is indeed extremely lightweight. Additionally, in Section 4.3, the author conducts experiments on efficiency with an ultra-long look-back window. The experimental results show that, even with an extremely long input length, TimeBase remains far more resource-efficient than DLinear, consistently maintaining very low resource consumption.

**Methods And Evaluation Criteria:**

The proposed method addresses the problem of efficient time-series forecasting on resource-constrained terminals, and the author has provided a comprehensive evaluation of the model's actual resource consumption and prediction performance.

**Other Comments Or Suggestions:**

The authors have chosen to bold the two best results in Table 2, but this does not indicate which method performs the best in terms of prediction accuracy. To improve the readability of the table, I recommend that the authors use different colors to highlight the first and second best results.

**Other Strengths And Weaknesses:**

Strengths:
1. Long-term time series forecasting is an important problem, and solving it efficiently has many real-world implications. This paper makes full use of the low-rank characteristics of long-term time series and innovatively proposes a lightweight time series forecasting model, effectively addressing this problem.
2. The efficiency validation of the model is thorough, especially the validation with extremely long inputs, which highlights TimeBase's lightweight advantage under extreme conditions.
3. The paper is well-structured and written, and the methodology is described clearly.

Weaknesses:
1. How TimeBase can be integrated into patch-based forecasting methods is not clearly and specifically described by the authors.
2. The authors set the batch size to 12 in Section 4.3. Is there a specific reason for this choice?
3. The authors only use two linear layers in the design of TimeBase, but from my understanding, both basis extraction and segment forecasting could potentially be achieved using transformers or CNNs. The authors should provide a detailed explanation of why they chose to use linear layers for the model design.

**Questions For Authors:**

Q1. I am curious—if the implementation of BasisExtract() is replaced with a more parameterized Attention mechanism instead of the linear layer, would the model’s prediction accuracy improve?

Q2. In the absence of prior knowledge or specific conditions, what methodologies can be employed to determine an appropriate P value for a given dataset?

**Relation To Broader Scientific Literature:**

Currently, there have been some explorations into lightweight time series forecasting [1][2], but these models have only managed to keep the parameter count above 1K. However, TimeBase has made a breakthrough by reducing the parameter count to below 0.4K while maintaining extremely low computational cost, further advancing the development of lightweight time series forecasting.

[1] SparseTSF: Modeling Long-term Time Series Forecasting with 1K Parameters, ICML 2024.

[2] FITS: Modeling Time Series with 10K Parameters, ICLR 2024.

**Theoretical Claims:**

In Appendix B, the author proves that the convergence lower bound of TimeBase is related to the orthogonality of the extracted temporal patterns, which provides a certain degree of credibility to the model's design.

---

> ### Author Rebuttal · Authors · 2025-03-30
>
> Dear Reviewer Pc2P,
>
> Thank you for taking the time to review our work and for your valuable feedback.
>
> ---
> >**D1. Discussion on Other Related Works**
>
> Thank you and `Reviewer 2K2o` for suggesting a comparison with related work and clarifying our distinctions and contributions. We provide an overall comparison in the table below, followed by a brief analysis and discussion.
>
> ||FITS|SparseTSF|TimeBase|
> |:-:|:-:|:-:|:-:|
> |Focused Domain|Frequency Domain|Time Domain|Time Domain|
> |Compression Strategy|Low-Pass Filtering|Downsampling|Basis Extraction|
> |Parameters|10.5K|1K|**0.39K**|
> |MACs|79.9M|12.71M|**2.77M**|
> |Time|35s|31.3s|**20.6s**|
>
> - **FITS** performs interpolation after low-pass filtering in frequency domain. FITS utilizes **frequency domain** feature, while TimeBase  only operates in  **time domain**, which is a fundamental difference between the two.
>
> - **SparseTSF** performs sparse prediction after local feature extraction, focusing on using TCN to **capture local features**. In contrast, TimeBase leverages the low-rank nature of long-term time series and enhances the encoding of key patterns through basis extraction. In terms of data efficiency, SparseTSF makes predictions **using all downsampled segments**, whereas TimeBase improves efficiency by **extracting only a small number of key basis components**.
>
> ---
> >**W1. Integration with Patch-Based Methods**
>
> TimeBase can serve as a lightweight complexity reducer for patch-based methods by replacing their feature extraction step with basis extraction and aggregation. Specifically for any patch-based models, after transforming the time series into **N** patches, TimeBase aggregates them into **R** core patches (**R ≪ N**). The model then performs forward computation based only on these **R** core patches, significantly reducing computational redundancy. For example, when integrated into PatchTST, TimeBase reduces the MACs from **14.17G to 1.58G** (**Table 7, Appendix C.2**).
>
> ---
> >**W2. Batch Size Selection (Section 4.3)**
>
> For efficiency analysis, we set the batch size to 12 for all models to ensure a fair comparison of efficiency metrics. This choice considers that an overly small batch size is impractical, while an excessively large batch size may cause some models to exceed memory limits. Following the comparison method used in FITS, we also set the batch size to 12 when testing the models' peak memory usage.
>
> ---
> >**W3. Why Use Linear Layers**
>
> TimeBase is designed to avoid unnecessary complexity while maintaining strong predictive performance. CNNs and Transformers are also effective in capturing sequential dependencies, but they introduce higher computational overhead. Additionally, empirical results from methods like DLinear and TimeMixer have demonstrated the effectiveness of linear models in time series forecasting. Therefore, our method leverages **linear basis extraction**, which efficiently captures key temporal patterns while significantly reducing computational cost.
>
> ---
> >**S1. Table 2 Formatting Improvement**
>
> Thank you for your suggestion. We will update Table 2 by using **different colors to highlight the best and second-best results**, improving readability.
>
> ---
> >**Q1: Replacing BasisExtract() with an Attention Mechanism**
>
> Thank you for your question. Here, we compare implementing BasisExtract() with Attention() VS the previous Linear(). The Attention-based BasisExtract() first applies a multi-head attention mechanism for global segment feature extraction, followed by a fully connected layer to map the features into a small set of basis components.
>
> The table below (720-input, 720-output on the Electricity dataset) shows their scale comparison, where the computational cost and parameter count of the Attention() version are 3-4 times higher than the previous implementation. The second table presents their prediction performance comparison (720-input, 720-output), indicating that on the Electricity and Traffic datasets, the Attention-based BasisExtract() achieves performance improvements. This suggests that increasing the parameter count enhances the expressiveness of the extracted basis components, leading to higher forecasting accuracy.
>
> BasisExtract($\cdot$)|MACs|Paras
> |:-:|:---:|:---:|
> **Linear($\cdot$)**|2.77M|0.39K
> **Attention($\cdot$)** |8.32M|0.99K
>
> BasisExtract($\cdot$)|Linear($\cdot$)||Attention($\cdot$)||
> |-|:-:|-|:-:|-|
> Metric|MSE|MAE|MSE|MAE
> Electricity|0.207|0.294|0.206|0.297
> Weather|0.309|0.331|0.324|0.351
> Traffic|0.456|0.298|0.451|0.294
> ETTm1|0.413|0.414|0.411|0.412
> ETTm2|0.352|0.380|0.348|0.384
> ETTh1|0.429|0.446|0.437|0.452
> ETTh2|0.400|0.448|0.398|0.443
>
> ---
> >**Q2: Selecting an Appropriate P Value Without Prior Knowledge**
>
> When no prior domain knowledge is available, **data-driven heuristics** such as Singular Spectrum Analysis (SSA) or Fourier Transform can help estimate dominant periods.

---

> > ### Comment · Reviewer_Pc2P · 2025-04-03
> >
> > I would like to thank the authors for their thoughtful responses to my comments. After thoroughly reviewing the rebuttal, I have no further concerns and continue to recommend acceptance of the paper.

---

### Official Review · Reviewer_JfQc · 2025-03-13

**Overall Recommendation:** 4

**Summary:**

This submission focuses on designing a simple yet efficient model to tackle complex forecasting problems and presents TimeBase, an ultra-lightweight network for long-term time series forecasting (LTSF). The experiments show that TimeBase has a small number of parameters, low computation and memory usage, and efficient CPU inference speed, which is significantly lower than the lightest DLinear. Moreover, TimeBase can not only function as a time-series forecaster but also serve as an efficient complexity reducer for patch-based methods.

**Claims And Evidence:**

Yes. The lightweight forecasting performance is validated in Sections 4.2 and 4.3, while its ability to serve as an efficient complexity reducer for patch-based methods is demonstrated in Section 4.4.

**Essential References Not Discussed:**

The discussion of related work is sufficient.

**Ethical Review Concerns:**

No ethical concerns.

**Experimental Designs Or Analyses:**

Yes. I think the experimental design is one of the highlights of this research. In particular, Section 4.3 demonstrates how TimeBase can reduce the complexity of PatchTST. Experimental results show that TimeBase reduces the computation of PatchTST to just 1/9 of its original MACs. This illustrates TimeBase's ability to significantly reduce the resource consumption of any patch-based model without sacrificing performance.

**Methods And Evaluation Criteria:**

Yes. The proposed method is valuable in LTSF and the effectiveness of the proposed solution is well-supported by extensive validation on real-world benchmark datasets.

**Other Comments Or Suggestions:**

1. The experimental results reported in Table 3 have inconsistent decimal precision, with some values rounded to two decimal places and others to three. The author should standardize this to three decimal places.

2. On page 5, lines 250-251, the author describes TimeMixer as an LLM-based method, but in fact, TimeMixer is not related to LLM. This should be corrected.

**Other Strengths And Weaknesses:**

Strengths:

1.The paper introduces and validates a novel approach for long-term time series forecasting by using temporal patterns for segment-level prediction rather than point-level forecasting.

2.TimeBase significantly reduces spatio-temporal complexity for any model. As revealed in the experiments, it can reduce the computation of PatchTST by at least 63%, greatly enhancing the efficiency of current time series forecasting models.

3.The paper conducts validation on 21 real-world datasets, covering data from different domains, which demonstrates the generalizability of TimeBase.

Weaknesses:

1.The proposed network is extremely lightweight. I suggest the authors explore real-world low-resource scenarios where TimeBase can be applied, to further deepen the motivation behind lightweight forecasting models.

2.All experiments are currently conducted with a 720 input setting. Considering that different models may have optimal input lengths, could the authors provide a comparison of predictions using a 336 input length to PatchTST? This could further strengthen the experimental results.

**Questions For Authors:**

I notice that the author made a correction to the 'drop_last' parameter in the code. What would happen if this correction had not been made?

**Relation To Broader Scientific Literature:**

I think the most significant contribution of this paper is that TimeBase can greatly reduce the spatio-temporal complexity of any model. As demonstrated in the experiments, it can reduce the computation of PatchTST by at least 63%, significantly improving the efficiency of current time series forecasting models.

**Theoretical Claims:**

Yes. In Appendix B, the author discusses the low-rank characteristics of long-term time series and confirms that the generalization lower bound of TimeBase is related to the learned time series basis, highlighting the importance of a high-quality basis vector space and the necessity of the orthogonal constraint.

---

> ### Author Rebuttal · Authors · 2025-03-28
>
> Dear Reviewer JfQc,
>
> Thank you for your interest in our paper and the constructive feedback you provide. We will address your questions point by point.
>
> ---
> >**W1. Real-world low-resource scenarios**
>
> Thank you for your suggestion! TimeBase can be applied in various resource-constrained environments, such as edge computing, mobile devices, and IoT-based forecasting. In these scenarios, the ability to handle large-scale time series data with minimal computational overhead makes TimeBase an ideal candidate.
>
> ---
> >**W2. Comparison with more input length**
>
> Thank you and `Reviewer 2K2o` for suggesting an analysis of different look-back window sizes effect to improve the experimental quality of our paper.
> In our original experiment, we choose a fixed length of 720 time steps to maintain consistency with FITS [1] and ensure a fair comparison. Additionally, we provide experimental **MAE comparisons with input lengths of [96, 336, 720]**, **all with a 720-step output length**, demonstrating that TimeBase maintains both its lightweight design and high prediction accuracy across different input lengths. Two interesting observations from this analysis are:
> - Most datasets, i.e., Electricity, ETTm1, ETTm2, show improved prediction performance as the input length increases.
> - For lightweight models such as TimeBase, SparseTSF, FITS, and DLinear, when the input length is only 96, none of them achieve good prediction performance on the Traffic dataset, but iTransformer and PatchTST could maintain prediction accuracy.
>
> | Model ||TimeBase|||SparseTSF|||FITS|||iTransformer|||DLinear|||PatchTST||
> |:---:|:---:|:---:|:---:|:---:|:---:|:---:|:---:|:---:|:---:|:---:|:---:|:---:|:---:|:---:|:---:|:---:|:---:|:---:|
> | Input length|96|336|720|96|336|720|96|336|720|96|336|720|96|336|720|96|336|720 |
> | Electricity|0.347|0.294 |0.294 |0.357|0.299 |0.300 |0.355|0.299 |0.302 |0.354|0.305 |0.301 |0.385|0.321 |0.309 |0.346|0.298 |0.299  |
> | Weather|0.321|0.336 |0.331 |0.328|0.343 |0.345 |0.329|0.343 |0.346 |0.324|0.336 |0.337 |0.363|0.363 |0.359 |0.313|0.335 |0.334  |
> | Traffic|0.384|0.310 |0.298 |0.387|0.305 |0.299 |0.386|0.308 |0.311 |0.305|0.354 |0.314 |0.394|0.378 |0.310 |0.311 |0.311 |0.289  |
> | ETTm1|0.448|0.421 |0.414 |0.449|0.417 |0.413 |0.449|0.416 |0.417 |0.457|0.436 |0.439 |0.452|0.424 |0.415 |0.44|0.425 |0.421  |
> | ETTm2|0.397|0.383 |0.380 |0.408|0.382 |0.380 |0.408|0.383 |0.380 |0.409|0.397 |0.407 |0.526|0.465 |0.433 |0.404|0.386 |0.386  |
> | ETTh1|0.461|0.439 |0.446 |0.464|0.446 |0.448 |0.459|0.453 |0.457 |0.501|0.507 |0.532 |0.53|0.516 |0.517 |0.476|0.471 |0.475  |
> | ETTh2|0.445|0.436 |0.448 |0.457|0.434 |0.425 |0.451|0.433 |0.423 |0.446|0.447 |0.470 |0.671|0.628 |0.609 |0.442|0.429 |0.434  |
>
> We appreciate your key suggestions for improving the quality of our paper presentation. We will include all results in the revised manuscript to provide more experimental analysis.
>
> ---
> >**S1. Decimal Precision in Table 3**
>
> Thank you for pointing this out. We will standardize the decimal precision to three decimal places for consistency in the revised manuscript.
>
> ---
> >**S2. TimeMixer Description Correction**
>
> We appreciate your attention to detail. We will correct the description of TimeMixer on page 5, lines 250-251, and clarify that it is not based on LLM. This will be updated in the revised manuscript.
>
> ---
> >**Q1. Impact of the 'drop_last' correction**
>
> The correction to the 'drop_last' parameter ensures that incomplete batches at the end of a sequence are excluded during testing. In some previous baselines, setting 'drop_last=True' with a very high batch size led to the discarding of difficult-to-predict samples, resulting in a significant improvement in prediction accuracy. This issue was later pointed out by FITS [1] and TFB [2], helping the time series forecasting field achieve better evaluation of predictive models. TimeBase addresses this bug to provide a fairer experimental comparison.
>
> ----
>
> [1] FITS: Modeling Time Series with 10k parameters. ICLR 2024.
>
> [2] TFB: towards comprehensive and fair benchmarking of time series forecasting methods. VLDB 2024.

---

### Official Review · Reviewer_2K2o · 2025-03-14

**Overall Recommendation:** 2

**Summary:**

This paper proposes an ultra-lightweight framework for long-term time series forecasting, TimeBase, which segments the time series, extracts basis components, and then performs forecasting. Furthermore, TimeBase can serve as a plug-and-play module to reduce the complexity of other patch-based models.

**Claims And Evidence:**

For the claim "TimeBase effectively captures essential patterns and interactions within the time series data, allowing it to retain or even slightly improve accuracy".  it does not clearly outline what essential patterns and interactions are being captured.

**Essential References Not Discussed:**

FITS and SparseTSF are two fundamental prior works that underpin this study, making it important to clarify their methodologies and highlight the distinctions and contributions of TimeBase.

**Experimental Designs Or Analyses:**

1. **Hyperparameter Search for Baselines:** Since the experiments are conducted on new datasets, was hyperparameter tuning performed for the baseline models? If so, can you provide details on the search space used for each baseline?

2. **Inference Time on CPU:** Why is inference time measured on a CPU? Would the performance gap between models be smaller if measured on a GPU? A comparison on GPU could provide a more balanced evaluation.

3. **Redundancy of Figure 4:** In the experiment "Efficiency in Ultra-long Look-back Window," Figure 4 appears redundant, as Theorem 3.1 already presents the parameter scale of TimeBase. Consider whether Figure 4 adds additional insights beyond what is already stated in the theorem.

4. **Impact of Look-back Window Size:** The study lacks experiments analyzing the effect of different look-back window sizes. Since the look-back window size significantly impacts performance, evaluating its influence would strengthen the analysis.

**Methods And Evaluation Criteria:**

1. **Is a fixed basis length P sufficient?** Time series data often exhibit multiple periodic patterns, yet the basis length P must be predefined, which may not fully capture these variations.
2. **Limited contribution:** The proposed method builds on SparseTSF and FITS, with its main novelty being basis extraction. This may constrain the paper’s overall contribution.
3. **Distinction between basis extraction and low-rank adaptation:** What differentiates basis extraction from low-rank adaptation in the context of this work?

**Other Comments Or Suggestions:**

Mentioned above.

**Other Strengths And Weaknesses:**

Mentioned above.

**Questions For Authors:**

1. In Figure 1(b), how is the singular value of the entire dataset calculated?
2. In Figure 2, why does the figure not include the latest baseline, SparseTSF?
3. The resolution of Figure 3 is low. Consider improving its clarity.
4. In line 251, TimeMixer is not an LLM-based method.
5. Can you provide the selected segment length P for all datasets?

**Relation To Broader Scientific Literature:**

Encouraging time series researchers to fully utilize sequential data and inspiring the development of backbone networks for pre-trained large-scale LTSF models.

**Theoretical Claims:**

1. Theorem 3.1 is similar to the theorem proposed in SparseTSF. However, it only compares the parameter scale with DLinear and lacks a comparison with SparseTSF, which limits the analysis of its relative efficiency and novelty.

---

> ### Author Rebuttal · Authors · 2025-03-28
>
> Dear Reviewer 2K2o,
>
> We greatly appreciate your constructive feedback and have made our best to address your concerns.
>
> ---
> >**C1. Sentence Explanation**
>
> |Term| Explain|
> |-|-|
> |Essential Patterns|The compact set of basis components extracted from redundant segments|
> |Interactions|Inter-correlations between extracted basis components by PatchTST|
>
> **This sentence (Page 6, Line 316-318) describes the plug-and-play integration of TimeBase into PatchTST.** TimeBase enhances the extraction of compact 'Essential Patterns' and enables a more effective 'Patch Interactions' for PatchTST, which leads to improved accuracy and reduced model complexity (as shown in Table 3 of main text) . We will refine the description to make it clearer and more understandable.
>
> ---
> >**M1. Fixed Basis Length**
>
> Most forecasting scenarios (e.g., traffic, electricity) feature a single dominant period or overlapping periods (e.g., a weekly period encompassing a daily period). Balancing data efficiency and accuracy, we set the shortest dominant period P to capture key temporal patterns.
>
> Besides, TimeBase can extract various-length basis using different P. In **Appendix C.6**, we show that
>
> $MSTimeBase = \sum_{i} TimeBase(X; P = p_i)$
>
> yields a 0.006 MSE improvement. The minor improvement suggests that the only capturing primary pattern by TimeBase is sufficient for lightweight forecasting.
>
> ---
> >**M2. Contribution of TimeBase**
>
> ||FITS|SparseTSF|TimeBase
> |-|-|-|-|
> Focused Features|Frequency Domain|Time Domain|Time Domain
> Compression Strategy |Low-Pass Filtering|Downsampling|Basis Extraction
> Paras|10.5K|1K|**0.39K**
> MACs|79.9M|12.71M|**2.77M**
> Time|35s |31.3s|**20.6s**
>
> TimeBase simplifies LTSF by extracting  basis components. In fact, it has distinct differences from FITS and SparseTSF  (Analysis is in `D.1 response to Reviewer PC2P`). Overall, **TimeBase’s Key Contributions** are three-fold,
> - **Modeling Techniques**: Propose basis extraction with orthogonal constraints to capture primary  patterns and avoid redundant computation from surge of time-series.
> - **Lightweight, Efficient and Effective Forecaster**: Achieve ultra lightweight but effective forecasting (0.39K Params, 2.77M Macs).
> - **Plug-and-Play Usability**:  Act as a complexity reducer for patch-based methods, cutting PatchTST’s MACs to 1/9 of its original one (e.g., 14.17G → 1.58G, **Tab.7, Appendix C.2**).
>
> ---
> >**M3. Basis Extraction and Low-Rank Adaptation**
>
> The key distinction between them lies in how to handle long-term dependencies in LTSF.
> - **Basis extraction** identifies a compact set of representative time patterns from historical data. Unlike low-rank approximation, TimeBase explicitly extracts key basis components and uses them as fundamental units for prediction.
> - **Low-rank adaptation**, in contrast, often relies on SVD to approximate data with a reduced-rank structure but does not explicitly extract  interpretable components.
>
> ---
> >**T1. Theoretical Supplement.**
>
> TimeBase|SparseTSF|DLinear|
> |-|-|-|
> $aT+bL+c$|$(TL)/P^2$|$2TL$|
> $a =(R+1)/P,b=R/P,c=R$
>
> TimeBase with **O(T+L)** scale is more efficient compared to SparseTSF/DLinear with **O(T*L)**  in LTSF. This will be added in revised version.
>
> ---
> >**E1. Baseline Settings**
>
> |Old Data|New Data|
> |-|-|
> |Default setting from official code|1. Search lr in [1e-4, 1e-3, 1e-2]|
> ||2. Fix & Unify other parameters: dim=64, enc_layers=2|
>
> ---
> >**E2. Inference Time**
>
> We provide CPU inference time as it is more practical for edge deployment. Thanks for your advice! We now include GPU (A100-80GB) inference times (ms) for 720-output Elec.
> |TimeBase|DLinear|Sparse|FITS|iTrans|
> |-|-|-|-|-|
> |**0.47**|0.58|0.65|1.09|3.3|
>
> ---
> >**E3. Function of Figure 4**
>
> Theorem 3.1 analyzes model scale theoretically. To intuitively verify TimeBase's efficiency, we visualize three computational metrics (Mem, Time, Paras) under ultra-long look-back windows in Figure 4.
>
> ---
> >**E4. Impact of Input Length**
>
> Thanks for your advice! We provide comparisons and analysis for input lengths [96, 336, 720] in the `W2 response to Reviewer JfQc`.
>
> ---
> >**D1. Discussion of FITS and SparseTSF**
>
> Thanks for your valuable advice. We have discussed the differences between TimeBase, FITS, and SparseTSF in the `D.1 response to Reviewer PC2P`, and we will add this discussion in the related work.
>
> ---
> >**Question**
>
> **Q1.** We apply a sliding window to segment the dataset into (num, patch) samples of 720 input length, perform SVD on each sample, sort the singular values, and then average them to derive the dataset’s typical singular value distribution.
>
> **Q2.** Figure 2 primarily highlights the comparison between TimeBase and DLinear. We have updated SparseTSF (1.0K Parameters, 125.2M Memory, 2.59ms CPU Inference Time, 12.71M MACs) in Figure 2, and you can check it in **our `original anonymous code link` in Abstract**. Thank you!
>
> **Q3-4.** Thank you very much! We will update them.
>
> **Q5.** P=4 (ETTm/Weather) | P=24 (Others)

---

### Official Review · Reviewer_n697 · 2025-03-14

**Overall Recommendation:** 4

**Summary:**

This manuscript focuses on improving efficiency in long-term time series forecasting. The proposed framework comprises only two linear layers, yet it achieves superior efficiency compared to recent state-of-the-art methods. Besides, it provides extensive theoretical analysis and plenty of experiments to demonstrate the effectiveness of the proposed method.

**Claims And Evidence:**

Yes. The claims are well-supported by theory analysis and extensive experiments.

**Essential References Not Discussed:**

No

**Experimental Designs Or Analyses:**

Yes, I have carefully reviewed the experimental design and analysis in the paper. I found that both the design and validation of the experiments are thorough, primarily in two aspects: (1) The paper uses 21 widely-used and publicly available real-world datasets, ensuring a sufficient and representative set of experiments. (2) The experimental validation is comprehensive, not only confirming the proposed method's high accuracy in predictions but also providing a detailed analysis of its efficiency, including model training time, memory usage, and computational complexity.

**Methods And Evaluation Criteria:**

Yes, the proposed method is validated on several real-world datasets. Its efficiency and accurate forecasting capabilities have a positive impact on long-term time series forecasting.

**Other Comments Or Suggestions:**

There is a minor typo to note, that Figure 7 is not referenced in Appendix. I recommend that the authors correct this in the revised version.

**Other Strengths And Weaknesses:**

**Strengths:**

S1. This paper proposes an extremely minimalistic model for long-term time series forecasting, significantly reducing the model's parameter count and computational complexity. It is an innovative and interesting contribution to the field of time series forecasting.

S2. The paper is well-structured, with a clear and concise methodology, accompanied by good figure illustrations and comprehensive tables, making it highly readable.

S3. The experiments are extensive, with validation conducted on a large number of real-world datasets, including 17 medium-scale and 4 large-scale datasets, thoroughly demonstrating the effectiveness of the proposed method.

S4. The plug-and-play nature of TimeBase significantly reduces the complexity for many patch-based methods, making it a promising area of research.

**Weaknesses:**

W1. Including an analysis that discusses the potential advantages of "Basis Orthogonal Restriction" could further enhance the contributions of TimeBase.

W2. Table 1 provides an efficiency comparison between TimeBase and other forecasting models, including Infer Time (CPU). The specific CPU model used in the experiments should be mentioned in the experimental setup.

W3. Section 3 describes the workflow for univariate time series data. Providing an explanation of how TimeBase can be applied to multivariate time series (MTS) data would help clarify and enhance the workflow description.

W4. I think the authors primarily focus on reshaping time series patterns of a single length. If a dataset contains multiple time series patterns of varying lengths, would TimeBase be well-suited to handle this?

**Questions For Authors:**

Q1. Does TimeBase support multiple temporal patterns? Such as Traffic could show daily and weekly periods.

Q2. I wonder how TimeBase can be applied to multivariate time series (MTS) data?

**Relation To Broader Scientific Literature:**

This paper focuses on the design of ultra-lightweight long-term time series forecasting models and achieves remarkably impressive results. I think this research is inherited from the base decomposition, such as [1].

[1] Bonizzi P, Karel J M H, Meste O, et al. Singular spectrum decomposition: A new method for time series decomposition[J]. Advances in Adaptive Data Analysis, 2014, 6(04): 1450011.

**Theoretical Claims:**

Yes. In Section 3.3, the authors provide a derivation of the parameter scale for the proposed method, and the derivation demonstrates that the model scale, as well as the input and output lengths, exhibit an extremely linear relationship, which is significantly smaller than that of the DLinear method.

---

> ### Author Rebuttal · Authors · 2025-03-28
>
> Dear Reviewer n697,
>
> Thank you for taking the time to review our paper and for your valuable suggestions to improve its quality.
>
> ---
> >**W1. Basis Orthogonal Restriction Analysis:**
>
> We appreciate your suggestion. From the perspective of the data space, the orthogonality of the basis vectors enhances its representation power, providing them ability to express as any vector in the data space through linear combination [1]. Therefore, the ts basis components should also be diverse and distinct, preventing the extraction of very single time-series patterns [2]. Based on this, we apply the Orthogonal Constraint. In the time-series space, each time series segment can be viewed as a composition of several orthogonal basis components. The goal of TimeBase is to transform the long-term forecasting task at the time-step level into a task of basis extraction and prediction at the period level, thus enabling an efficient time-series forecasting model (0.39K, 1000 times smaller than DLinear). In the revised version, we will add a detailed analysis discussing the potential advantages of "Basis Orthogonal Restriction."
>
> ---
> >**W2. CPU Detail:**
>
> Thank you for pointing this out.  The CPU used in our experiments is an Intel Xeon E5-2609 v4 CPU (16 cores, 2 sockets, 1.7 GHz base frequency), and we will update the manuscript accordingly.
>
> ---
> >**W3. TimeBase for Multivariate Time Series:**
>
> Thanks for your question. TimeBase does not consider relationships between variables; instead, it transforms the multivariate time series forecasting task into multiple univariate forecasting tasks. We have  explained this in the original manuscript on **Page 3, lines 150-156**, “ Most existing multivariate time series are homogeneous, meaning that each sequence within the dataset exhibits similar periodicity [3]. This characteristic allows them to be organized as a unified multivariate time series. Based on this property, we employ the Channel Independence [4] to simplify the forecasting of MTS data into separate univariate forecasting tasks.”
>
> ---
> >**W4&Q1. Handling Multiple Time Series Patterns:**
>
> Most forecasting scenarios (e.g., traffic, electricity) exhibit a dominant period or overlapping periods (e.g., a weekly period encompassing a daily period). Therefore, we set the shortest period as the fixed P.  For datasets with multiple periods, TimeBase can extract different basis components using various P. In **Appendix C.6**, we demonstrate that
>
> $\text{MSTimeBase} = \sum_{i} \text{TimeBase}(X; P = p_i)$
>
> could achieve a 0.006 MSE improvement on the traffic dataset. The marginal improvement suggests that the primary patterns could been effectively captured by single&primary basis length .
>
> ---
> >**C1. Minor Typo**
>
> We acknowledge the typo regarding Figure 7 not being referenced in the Appendix. We will correct this and ensure that Figure 7 is properly referenced in the revised manuscript.
>
> ---
> >**Question**
>
> **Q2.** TimeBase can be applied to MTS data by independently extracting basis components from each time series in the multivariate dataset. Each time series is processed to capture its key temporal patterns, and these components are then aggregated to form a unified forecast.
>
> ---
>
> [1]  Time series with periodic structure. Biometrika 1967.
>
> [2] Decomposition principle for linear programs. Operations research 1960.
>
> [3] TimesNet: Temporal 2D-Variation Modeling for General Time Series Analysis, ICLR 2023.
>
> [4] A Time Series is Worth 64 Words: Long-term Forecasting with Transformers, ICLR 2023.

---

### Decision · Program_Chairs · 2025-05-01

**Decision:**

Accept (spotlight poster)

**Comment:**

This paper introduces a new model for time series forecasting. I am pleasantly surprised by the strong performance, especially given the simplicity of the proposed model. The trade-off between efficiency and accuracy is quite impressive. In addition, the paper offers useful insights that are valuable for the design of time series forecasting models.

Overall, the reviews are very positive, and the authors have addressed all major concerns during the rebuttal phase. Notably, it appears that most of Reviewer 2K2o's concerns have been resolved as well. I also recall seeing this paper previously at ICLR 2025, and it seems that the authors have addressed the earlier feedback in this revised version.

In addition to the reviewers' comments, my main suggestion is to improve the clarity and presentation of the figures. Figures 1 and 2 are currently difficult to parse and offer limited value in their present form. Figure 3 contains useful information but is quite crowded, adding more space could help readability. The in-plot elements of Figure 4 are also hard to parse. I recommend revisiting all figures for the camera-ready version, possibly moving one or two figures from the results section to the appendix to make room for cleaner, more informative visuals.

That said, I believe this paper is a solid contribution to ICML, and I recommend accepting it.